# Anytime-Valid Inference for Online Ranking of Large Language Models

**Runzhe Gu** [1 2]  **Wenguang Sun** [2]  **Bowen Gang** [3]  **Xintao Xia** [2]

## Abstract

Online evaluation of large language models increasingly relies on sequentially collected pairwise preferences, enabling human-aligned assessment and continuous data collection until closely performing models can be reliably distinguished. However, adaptive sampling and continuous monitoring invalidate classical fixed-sample inference, rendering existing ranking procedures largely heuristic. We propose SERPANT (Sequential E-value Ranking and Pruning via Adaptive Null Testing), a principled framework for online LLM ranking with anytime-valid guarantees. SERPANT formulates model comparison as a collection of pairwise hypothesis tests and constructs e-processes to ensure family-wise error rate control at any monitoring time. Anytime validity provides a theoretical justification for early stopping, enabling substantial cost savings from expensive human annotation. To improve efficiency, we introduce a novel tournament-based sampling strategy that adaptively selects comparisons based on past outcomes. The proposed framework is also extended to top-k model identification.

## 1. Introduction

Large language models (LLMs) have become the central engines of modern artificial intelligence, achieving impressive performance across a wide range of tasks (Vaswani et al., 2017; Jumper et al., 2021; Alayrac et al., 2022; Kojima et al., 2022; Radford et al., 2023; Kasneci et al., 2023; Kirillov et al., 2023). The rapid proliferation of LLMs has created an urgent need for reliable, scalable, and cost-effective methods for evaluating and comparing their performance. As organizations and researchers select models for

deployment–whether in conversational agents, content generation, or complex reasoning tasks–the question *"which model is best?"* has shifted from a theoretical inquiry to a pressing practical decision with significant implications for cost, safety, and effectiveness. However, existing evaluation practices are largely *heuristic*, relying primarily on predefined statistics measuring reasoning ability, truthfulness, safety, and domain-specific knowledge (Kiela et al., 2021; Peng et al., 2024). Such evaluations rely heavily on *benchmark datasets*, which typically measure limited skill aspects and are often not representative of human judgments and preferences. Meanwhile, recent LLMs achieve similar scores on offline benchmarks (Chiang et al., 2024). Their fixed sample size fundamentally limits discriminatory power for distinguishing closely performing models.

**Online Ranking and Sequential Design.** To address these limitations, the community is increasingly turning to dynamic evaluation through pairwise comparisons based on human feedback, as exemplified by platforms like Chatbot Arena (Chiang et al., 2024). This paradigm captures subjective, holistic judgments that are often more aligned with real-world utility. While high-quality human feedback has played a central role in LLM development (Christiano et al., 2017; Ouyang et al., 2022; Bai et al., 2022; Glaese et al., 2022), its collection is costly and time-consuming, leading to two key challenges for efficient online evaluation.

First, we require a sequential experimental design to maximize information gain under a constrained budget. Instead of pre-committing to a fixed number of comparisons for all model pairs, a sequential strategy allows us to continuously monitor evidence and stop comparing a pair once a confident preference is established. This enables the reallocation of valuable human effort to more uncertain or informative comparisons, dramatically improving sampling efficiency.

Second, the statistical validity of inferences under continuous monitoring is often undermined by "data peeking", where results are repeatedly examined with the intent to stop early. Such adaptive behavior, if unaccounted for, can inflate Type I error rates and invalidate standard inference. This creates a critical need for inferential frameworks with *anytime-valid guarantees* (Howard et al., 2021; Johari et al., 2022; Grünwald et al., 2024), ensuring that confidence statements and error control hold not only at a prespecified sam-

[1]School of Mathematical Sciences, Zhejiang University, Zhejiang, China [2]Center for Data Science, Zhejiang University, Zhejiang, China [3]Department of Statistics and Data Science, School of Management, Fudan University, Shanghai, China. Correspondence to: Xintao Xia <xintaox@zju.edu.cn>.

*Proceedings of the 43rd International Conference on Machine Learning*, Seoul, South Korea. PMLR 306, 2026. Copyright 2026 by the author(s).

ple size, but also at any arbitrary, data-dependent stopping time during the evaluation process. Moreover, uncertainty quantification for rankings, particularly for recovering top-performing models, remains underexplored in sequential settings.

**The SERPANT Algorithm.** We introduce SERPANT (Sequential E-value Ranking and Pruning via Adaptive Null Testing), a novel framework that addresses the above challenges by unifying anytime-valid inference with adaptive sequential design. Our approach formulates LLM ranking as a multiple-comparison problem with adaptively sampled pairwise comparisons. At its core, SERPANT operates as an online tournament. It dynamically maintains an active set of unresolved pairwise hypotheses and employs a novel *Tournament Sampling* strategy to prioritize comparisons that are either highly uncertain or exhibit strong signals. This adaptive allocation of evaluation resources accelerates the identification of dominance relations. An implementation of SERPANT is available at `https://github.com/ranzer30/serpant_python`.

## 1.1. Our Contributions

We summarize our contributions as follows:

- **Anytime-Valid Ranking Inference**: We develop the first sequential ranking procedure for LLMs that guarantees family-wise error rate (FWER) at any stopping time. Our method produces dynamically updated, uncertainty-aware partial rankings that remain valid throughout continuous monitoring.

- **Efficient Sequential Design**: We introduce Tournament Sampling, an adaptive pair-selection strategy that prioritizes informative comparisons. This design substantially accelerates the discovery of true performance differences while seamlessly integrating with the inferential framework without breaking validity guarantees.

- **Top-$k$ Identification Framework**: We extend the pairwise testing framework to construct anytime-valid confidence sets for the top-$k$ models, directly addressing the practical need to identify an elite subset rather than a complete ranking. This provides a principled, uncertainty-quantified alternative to heuristic leaderboards.

## 1.2. Related Works

**LLM Benchmarks.** Evaluation tasks such as logical, mathematical, and multi-hop reasoning are primarily assessed through benchmarks bAbI (Weston et al., 2015), $\alpha$-NLI (Bhagavatula et al., 2020), GSM8K (Cobbe et al., 2021), and HotpotQA (Yang et al., 2018). Another major component of evaluation concerns truthfulness and safety outputs

such as TruthfulQA (Lin et al., 2022), ToxicChat (Lin et al., 2023), and SafetyBench (Scherrer et al., 2023). Techniques such as HaluEval 2.0 (Jiang et al., 2024) benchmarks have been developed for hallucination detection. Domain knowledge is further assessed through field-specific benchmarks in finance (Wu et al., 2023), law (Blair-Stanek et al., 2023), psychology (Demszky et al., 2023), and medicine (Agrawal et al., 2022).

**Ranking.** Prior work on statistical inference for ranking models has established model-based asymptotic guarantees for latent preference parameters and recovering rankings under various comparison regimes (Rao & Kupper, 1967; Han et al., 2020; Liu et al., 2023). Extensions to multiway choice settings further develop ranking inference and top-k identification procedures (Fan et al., 2025; 2026). Additional studies incorporate covariates into comparison models and analyze uncertainty in nonparametric settings (Fan et al., 2024; Wang et al., 2025). Finite-sample approaches remain limited, with recent work providing marginal confidence sets (Bazylik et al., 2025) and resampling methods (Xie & Wang, 2024). However, these methods do not provide anytime-valid guarantees for online evaluation.

**Anytime-valid Inference.** Sequential testing (Wald, 1945; Siegmund, 1985) is a well-established discipline focused on making decisions based on sequentially collected data. Since naive peeking can greatly inflate Type-I error (Armitage et al., 2018), methods that remain valid under continuous monitoring and arbitrary stopping times are highlighted (Robbins & Siegmund, 1974). Driven by modern online experimentation, recent work has established frameworks for anytime-valid p-values and e-processes (Johari et al., 2017; 2022; Howard et al., 2021; Grünwald et al., 2024; Russac et al., 2021; Casgrain et al., 2024). Recent developments in sequential error control further improve practical reliability (Jamieson & Jain, 2018; Ham et al., 2023; Lindon & Malek, 2022; Dalal et al., 2024).

## 2. Statistical Framework for Anytime-Valid LLM Ranking

We consider an online evaluation setting in which human volunteers provide pairwise preference judgments over responses generated by LLMs. Let $\{l_i(\cdot)\}_{i=1}^{M}$ denote a collection of $M$ candidate models to be ranked.

**Rank as pairwise comparison.** Rather than directly modeling a global ranking, we represent model performance through an unknown *pairwise comparison matrix*

$$C = (C_{j,k})_{1 \leq j \neq k \leq M},$$

where $C_{j,k}$ encodes the underlying preference between models $l_j$ and $l_k$. Specifically, $C_{j,k} = 1$ indicates that model $l_j$ is preferred to $l_k$, while $C_{j,k} = -1$ indicates the opposite

preference. We impose the following structural assumptions on the comparison matrix:

- *No ties.* For all $j \neq k$, $C_{j,k} \in \{1, -1\}$.

- *Transitivity.* If $C_{j,k} = C_{k,m} = 1$, then $C_{j,m} = 1$.

Under these assumptions, the comparison matrix induces a total order over models.

*Remark* 2.1 (Role of transitivity). Transitivity is used only to improve efficiency through pruning. If the preference relation is transitive, discovered comparisons can be propagated. For example, $l_j \succ l_k$ and $l_k \succ l_m$ imply $l_j \succ l_m$, eliminating the need to sample the pair $(l_j, l_m)$. This assumption is natural in mathematics, coding, or factual QA. For subjective settings where preferences may be cyclic, the pruning step can be removed. The unpruned procedure still performs **anytime-valid** sequential tests and controls the FWER. Thus, transitivity affects efficiency, but not the validity. See Appendix B.3 for a complete discussion.

For each ordered model pair $(j, k)$, we consider the directional hypothesis

$$H_0^{(j,k)} : C_{k,j} = 1,$$

corresponding to the null that model $l_j$ is not preferred to model $l_k$. Rejecting $H_0^{(j,k)}$ provides statistical evidence supporting the strict preference $l_j \succ l_k$ and the model pair $(l_j, l_k)$ is removed from future comparisons. Let $\widehat{C}^t$ denote the estimated comparison matrix at time $t$. At the beginning, we set $\widehat{C}_{j,k}^0 = 0$ for $1 \leq j \neq k \leq M$, where a zero entry indicates that the corresponding pairwise preference has not yet been determined.

**Adaptive sampling and sequential testing.** At each discrete time $t = 1, 2, \ldots$, a prompt $x_t$ is submitted, and a model pair $(l_j, l_k)$ satisfying $\widehat{C}_{j,k}^{t-1} = 0$ is selected according to a *sequential sampling design*. The selected models generate responses

$$y_t^j(x_t) \sim l_j(x_t), \qquad y_t^k(x_t) \sim l_k(x_t),$$

which are then presented to a human evaluator for comparison. The observed outcome is recorded as

$$z_{j,k}^t = \begin{cases} 1, & \text{if } y_t^j \text{ is preferred to } y_t^k, \\ 0, & \text{otherwise.} \end{cases}$$

We define the prompt-marginal preference probability

$$p_{j,k} := \mathbb{E}_{X \sim \mathcal{P}_X} \left[ \mathbb{P}(y^j(X) \succ y^k(X) \mid X) \right],$$

where the prompts $\{x_t\}_{t \geq 1}$ are i.i.d. draws from an underlying distribution $\mathcal{P}_X$. Conditional on a prompt $X$, the model responses satisfy $y^j(X) \sim l_j(X)$ and $y^k(X) \sim l_k(X)$.

Then, $C_{j,k} = 1$ if $p_{j,k} > 1/2$, and $C_{j,k} = -1$ otherwise. Despite adaptive pair selection, the outcomes $z_{j,k}^t$ satisfy $\mathbb{P}(z_{j,k}^t = 1) = p_{j,k}$ and thus constitute evidence for the hypothesis $H_0^{(j,k)}$. As observations accumulate sequentially, hypotheses may be rejected over time.

For each unordered model pair $\{l_j, l_k\}$, the two directional hypotheses are tested jointly; once one is selected, the other is discarded. Thus, we set $\widehat{C}_{j,k}^t = 1$ and $\widehat{C}_{k,j}^t = -1$ if the hypothesis $H_0^{(k,j)}$ has been rejected at or before time $t$. When a rejection occurs, we further update the matrix $\widehat{C}^t$ by the transitivity assumption. Specifically, if $\widehat{C}_{j,k}^t = 1$ and $\widehat{C}_{k,m}^t = 1$, then we set $\widehat{C}_{j,m}^t = 1$ and $\widehat{C}_{m,j}^t = -1$.

*Remark* 2.2 (Ties and unresolved pairs). Our framework can accommodate ties, and the no-ties assumption is imposed only for clarity of presentation. A true tie between two models $l_j$ and $l_k$ corresponds to the boundary case $p_{j,k} = 1/2$. This case is included in the null hypothesis, where $H_0^{(j,k)} : p_{j,k} \leq 1/2$ versus $H_1^{(j,k)} : p_{j,k} > 1/2$, and therefore does not invalidate the e-process construction or the FWER guarantee. If two models are truly tied, the corresponding e-process may not cross the rejection threshold, so the pair remains unresolved. This is the intended behavior: when the evidence is insufficient to establish a strict preference, the procedure treats the models as statistically indistinguishable rather than forcing an arbitrary ordering.

**Error rate and anytime validity.** We define the false selection set at time $t$ as

$$\mathcal{V}_t = \left\{ (j, k) : 1 \leq j \neq k \leq M, \ \widehat{C}_{j,k}^t \neq 0, \ \widehat{C}_{j,k}^t \neq C_{j,k} \right\}$$

namely, the subset of selected hypotheses corresponding to incorrect pairwise preferences. The family-wise error rate (FWER) at time $t$ is given by

$$\text{FWER}(t) = \mathbb{P}(\mathcal{V}_t \neq \emptyset).$$

In a sequential setting, inference may be monitored continuously and terminated at an arbitrary time. Let $\tau$ denote any (possibly random and data-dependent) stopping time with respect to the filtration generated by the observed comparisons. An *anytime-valid FWER* guarantee at level $\alpha$ requires

$$\sup_\tau \mathbb{P}(\mathcal{V}_\tau \neq \emptyset) \leq \alpha.$$

The set of selected pairwise preferences $\widehat{C}^t$ induces a *directed acyclic graph* over the models, thereby defining a partial order (see Appendix B for details). As additional comparisons are collected, this partial order becomes increasingly informative and eventually yields a total ranking.

# 3. Anytime-Valid LLM Ranking Inference

## 3.1. E-Processes for Pairwise Preference Testing

Online ranking of LLMs requires continuous monitoring of adaptively collected data. For each hypothesis $H_0^{(j,k)}$, we conduct sequential testing using the framework of *e-processes*, which provides anytime-valid guarantees. An e-process $(E_t)_{t \geq 0}$ is a sequence of nonnegative random variables satisfying $\mathbb{E}[E_\tau] \leq 1$ under the null hypothesis, for any stopping time $\tau$. As a result, thresholding an e-process yields a valid sequential test under continuous monitoring.

**Baseline e-process.** We first consider a setting in which the covariate is unavailable and develop a construction based on observed binary preference outcomes. When the context is clear, we suppress the subscripts $(j,k)$ to simplify notation. At time $t$, let $T_t \subset \{1, \ldots, t\}$ denote the set of time indices, with size $n_t$, at which models $l_j$ and $l_k$ are selected for comparison. Define $S_t = \sum_{i \in T_t} z_{j,k}^i$ as the number of times model $l_j$ is preferred to model $l_k$ up to time $t$. The likelihood is $L(p; S_t, t) = p^{S_t}(1-p)^{n_t - S_t}$. To test the hypothesis $H_0^{(k,j)}$, we construct a mixture likelihood ratio e-process by

$$e_{j,k}^t = \frac{\int_{1/2}^1 L(p; S_t, t) \, g(p) \, dp}{\sup_{p \in [0,1/2]} L(p; S_t, t)}, \quad (1)$$

where $g(p)$ is a weighting distribution over the alternative parameter space.

**Proposition 3.1.** *Under the null hypothesis $H_0^{(j,k)}$, the process $(e_{j,k}^t)_{t \geq 0}$ defined by (1) is a nonnegative e-process with respect to the filtration generated by the observed comparison outcomes up to time $t$.*

Proposition 3.1 establishes the foundation of our construction. The e-process ensures that, under the null hypothesis, the expected value of $e_{j,k}^t$ remains bounded by one at all stopping times. This property is essential for enabling valid inference under optional stopping and adaptive sampling. Under the alternative, although both terms in (1) vanish asymptotically, the denominator decays at a strictly faster exponential rate, yielding exponential growth of the e-process and hence detection power. A convenient and effective choice is the uniform weighting $g(p) = 2 \cdot \mathbb{I}(p \in (1/2, 1])$, which yields the closed-form expression

$$e_{j,k}^t = \frac{2 \int_{1/2}^1 p^{S_t}(1-p)^{n_t - S_t} \, dp}{(\hat{p}_{j,k}^t)^{S_t}(1 - \hat{p}_{j,k}^t)^{n_t - S_t}}, \quad \hat{p}_{j,k}^t = \min\left\{\frac{S_t}{n_t}, \frac{1}{2}\right\}. \quad (2)$$

This baseline e-process does not rely on additional modeling assumptions and serves as a natural choice when prompt information is unavailable or difficult to incorporate.

**Covariate-assisted e-processes.** In many LLM evaluation settings, each comparison is accompanied by observable covariates $x_i \in \mathcal{X}$, for $i = 1, \ldots, t$ (e.g., prompt features). We incorporate covariate information through the alternative mixture. Specifically, we adopt a covariate-augmented Bradley–Terry (BT) model, a widely used framework in preference learning and LLM alignment:

$$\mathbb{P}(z_{j,k}^i = 1 \mid x_i) = \sigma(x_i^\top \theta_{(j,k)}), \sigma(t) = \{1 + \exp(-t)\}^{-1}, \quad (3)$$

where $\theta_{(j,k)}$ denotes a latent parameter vector associated with the ordered pair $(j, k)$. The BT model in (3) is used solely for power enhancement; all validity guarantees hold regardless of model misspecification. Let $\hat{\theta}_{(j,k)}^{(t-1)}$ denote the estimator based on history data up to time $t - 1$, and define the model-predicted preference probability $\hat{p}_t = \sum_{i \in T_{t-1}} \sigma(x_i^\top \hat{\theta}_{(j,k)}^{(t-1)})/n_{t-1}$. The quantity $\hat{p}_t$ reflects the model-predicted preference of $j$ over $k$ and is used to shape a prior. We construct a time-varying prior $\hat{g}_t(\cdot)$ supported on $[1/2, 1]$ by shifting a Beta distribution. Specifically, let $\hat{g}_t(\cdot)$ denote the density of $1/2 + U_t/2$, where $U_t \sim \text{Beta}(\alpha_t, \beta_t)$. To align the prior mean with $\hat{p}_t$, we impose $\alpha_t/(\alpha_t + \beta_t) = 2(\hat{p}_t - 1/2)$, for example by choosing $\alpha_t = 1 + \max\{\hat{p}_t - 1/2, 0\}$ and $\beta_t = 1$. This construction yields a uniform prior when $\hat{p}_t \approx 1/2$, and a more concentrated prior when the Bradley–Terry model provides stronger evidence against the null. The resulting covariate-assisted e-process is

$$\tilde{e}_{j,k}^t = \frac{\prod_{i \in T_t} \int_{1/2}^1 p^{z_{j,k}^i}(1-p)^{1-z_{j,k}^i} \, \hat{g}_i(p) \, dp}{\sup_{p \in [0,1/2]} L(p; S_t, t)}. \quad (4)$$

The resulting process $(\tilde{e}_{j,k}^t)_{t \geq 1}$ remains a valid e-process while incorporating covariate information through the BT model to enhance statistical power. The theoretical details are provided in Appendix A.

## 3.2. The SERPANT Algorithm

Based on the e-process, we introduce the **S**equential **E**-value **R**anking and **P**runing via **A**daptive **N**ull **T**esting (SERPANT) algorithm. At time $t$, the algorithm adaptively selects a pair of models whose relative ranking has *not been determined*, according to a *sequential sampling design*, and updates their corresponding e-values as new comparison data arrive. At each time step, each hypothesis is monitored via an e-process, ensuring statistical validity under continuous monitoring and data-dependent stopping rules. A hypothesis is rejected once its e-value exceeds a preset threshold $K$. We set $K = m(m-1)/\alpha$ corresponding to a *Bonferroni-style correction* for controlling the FWER over the $m(m-1)$ pairwise comparisons. This guarantee is agnostic to the sampling strategy and stopping rule.

The complete procedure is summarized in Algorithm 1. Beyond direct rejections, SERPANT leverages the transitive structure in ranking problems. Specifically, if the algorithm

**Algorithm 1** SERPANT: Anytime-Valid Online Ranking

**Require:** Models $\{l_1, \dots, l_m\}$; FWER level $\alpha \in (0,1)$.
1: Initialize $t \leftarrow 0$, $K \leftarrow \frac{m(m-1)}{\alpha}$, $e^0_{j,k} \leftarrow 1$, and $\widehat{C}^0_{j,k} \leftarrow 0$ for all $j \neq k$.
2: **while** data arrive and $\widehat{C}^t_{j,k} = 0$ for some $j \neq k$: **do**
3:    $t \leftarrow t + 1$
4:    Select a model pair $(l_j, l_k)$ from the candidate set $\{(j,k) : \widehat{C}^{t-1}_{j,k} = 0\}$ according to a sampling design.
5:    Observe the comparison outcome $z^t_{j,k}$ and update the corresponding e-values $e^t_{j,k}$ and $e^t_{k,j}$ using (1) or (4).
6:    **if** $e^t_{j,k} \geq K$ **then**
7:       Reject $H_0^{(j,k)}$ and update $\widehat{C}^t_{j,k} = -1$, $\widehat{C}^t_{k,j} = 1$.
8:    **else if** $e^t_{k,j} \geq K$ **then**
9:       Reject $H_0^{(k,j)}$ and update $\widehat{C}^t_{j,k} = 1$, $\widehat{C}^t_{k,j} = -1$.
10:   **end if**
11:   Prune by transitivity: for any $h, l, m$, if $\widehat{C}^t_{h,l} = 1$ and $\widehat{C}^t_{l,m} = 1$, then set $\widehat{C}^t_{h,m} = 1$ and $\widehat{C}^t_{m,h} = -1$.
12: **end while**
**Ensure:** Estimated comparison matrix $\widehat{C}^t$.

has already rejected $H_0^{(h,l)}$ and $H_0^{(l,m)}$, yielding the conclusions $l_l \succ l_h$ and $l_m \succ l_l$, transitivity implies $l_m \succ l_h$ without requiring additional observations. The *Prune by transitivity* step formalizes this principle by automatically rejecting hypotheses whenever such a chain exists. This step improves both statistical and computational efficiency.

**Theorem 3.2.** *The SERPANT procedure (Algorithm 1) controls the Family-Wise Error Rate (FWER) at level $\alpha$ at any stopping time $T$.*

Theorem 3.2 shows that SERPANT maintains strong error control under continuous monitoring and data-dependent stopping, which directly implies the reliability of the rankings as shown in Corollary 3.3.

**Corollary 3.3.** *The estimated ranking produced by the SERPANT algorithm satisfies the following condition at all times*

$$\mathbb{P}\Big(\exists j \neq k \text{ such that } l_j \prec l_k \text{ but } H_0^{(k,j)} \text{ is rejected}\Big) \leq \alpha.$$
$$(5)$$

# 4. Power Enhancement through Tournament-Based Aggregation

Because data collection remains costly, our objective is to identify as many reliable pairwise orderings as possible given a fixed evaluation budget. Uniform random sampling over model pairs can be inefficient, as it may waste resources on comparisons with weak performance differences. To substantially improve the statistical power of SERPANT while preserving its anytime-valid guarantees, we introduce *Tournament Sampling*, an adaptive pairwise sampling strat-

egy inspired by successive elimination and ranking bandits (Jamieson & Nowak, 2014; Chen et al., 2016). In each round, the algorithm acts as an active learning scheduler, dynamically selecting a batch of model pairs for comparison based on their current statistical states. The sampling order is explicitly guided by two complementary considerations: uncertainty in the pairwise outcome and signal strength indicating a potential performance gap. Concretely, using the current win proportion $\hat{p}^t_{j,k}$ defined in (2), we assign each ordered pair $(j,k)$ a priority score $Q^t(j,k)$. In the next evaluation round, the top $q$ pairs are selected for comparison in descending order by their priority scores, where $q$ is a user-specified parameter. The priority score $Q^t(j,k)$ is defined as a weighted combination of two components:

$$Q^t(j,k) = w_1 \cdot U^t(j,k) + w_2 \cdot S^t(j,k),$$

where $w_1 + w_2 = 1$.

- Uncertainty: $U^t(j,k) = 1 - 2|\hat{p}^t_{j,k} - 0.5|$ rewards pairs with win rates close to 0.5.

- Signal Strength: $S^t(j,k) = 2|\hat{p}^t_{j,k} - 0.5|$ favors pairs showing strong evidence of deviation from 0.5.

*Remark* 4.1. When the weight parameter is set so that all pairs receive the same priority score (e.g., $w_1 = w_2 = 0.5$), the selection will degenerate to uniform random sampling.

This linear weighting balances exploration and exploitation by combining uncertainty-driven and signal-driven sampling. The weights are flexible and task-dependent, with empirical behavior examined in Appendix C, and their choice does not affect the anytime-validity guarantees of the proposed procedure. In addition, we adopt a cyclic schedule to balance exploration and exploitation. After several exploitation rounds, an exploration round is triggered in which the algorithm targets "cold" pairs, namely those whose sample counts fall below the $1 - l$ quantile, where $l$ is user-defined. This mechanism prevents the procedure from becoming trapped in local decisions and maintains sensitivity to effects that may initially appear weak. Complete algorithmic details are provided in Appendix B.

# 5. Anytime-Valid Confidence Sets for the Top-k Models

In large-scale model evaluation, the objective is often not to recover a complete ranking of all models, but rather to reliably identify a small subset of the best-performing ones. The *top-k problem* aims to construct a set of models that contains all truly top-k models with high probability. Let $r_j$ denote the unknown rank of model $l_j$, where smaller values correspond to better performance. Our goal is to construct a data-dependent set $\mathcal{C}_k$ satisfying

$$\mathbb{P}\left(\{l_j \mid r_j \leq k\} \subseteq \mathcal{C}_k\right) \geq 1 - \alpha.$$

Our existing pairwise hypothesis testing framework provides all the necessary components to solve this problem. The key insight comes from (Bazylik et al., 2025), which demonstrates that confidence sets for the ranks of individual items can be constructed from a collection of pairwise tests.

At any time $t$, the estimated comparison matrix $\widehat{C}^t$ encodes information about the relative ordering of the models. This information can be used to derive lower and upper bounds on each model's true rank. The rank of model $l_j$, denoted by $r_j$, is defined as $r_j = 1 + \sum_{i \neq j} \mathbb{I}(l_i \succ l_j)$. Based on the rejected pairwise hypotheses, we define two quantities based on our tests:

- The number of models confirmed to be better than $l_j$:
  $N_j^+(t) = |\{i : \widehat{C}_{i,j}^t = 1\}|$.

- The number of models confirmed to be worse than $l_j$:
  $N_j^-(t) = |\{i : \widehat{C}_{i,j}^t = -1\}|$.

A valid lower bound on the rank of $l_j$ is $L_j(t) = 1 + N_j^+(t)$. Because SERPANT controls FWER, every model counted in $N_j^+(t)$ is, with high probability, truly better than $l_j$. An upper bound on the rank follows from a complementary argument. Among the $m - 1$ remaining models, $N_j^-(t)$ have been confirmed to be worse than $l_j$. Hence, at most $m - 1 - N_j^-(t)$ models still can outperform $l_j$, yielding $U_j(t) = m - N_j^-(t)$. Together, a $(1 - \alpha)$ confidence set for the rank of model $l_j$ is:

$$\text{CS}_j(t) = [L_j(t), U_j(t)] = [1 + N_j^+(t), m - N_j^-(t)].$$

The validity of this confidence set follows directly from the FWER guarantee of the SERPANT algorithm. Consequently, $\mathbb{P}(\forall j, r_j \in \text{CS}_j(t)) \geq 1 - \alpha$.

With confidence sets for each model's rank, we can now define a confidence set for the identities of the top-$k$ models. A model $l_j$ is a plausible candidate for being in the top-$k$ if its confidence set for its rank, $\text{CS}_j(t)$, has a non-empty intersection with the interval $[1, k]$. Using the rank confidence sets, we define a confidence set for the identities of the top-$k$ models as

$$\mathcal{C}_k(t) = \{\, l_j : \text{CS}_j(t) \cap [1, k] \neq \emptyset \,\}. \tag{6}$$

Equivalently, this set contains all models whose lower rank bound satisfies $L_j(t) \leq k$.

The following theorem ensures that, at any time, the reported top-$k$ candidate set (6) contains all truly top-$k$ models with high probability. This result enables researchers to make early, reliable decisions about leading models while data collection is ongoing.

**Theorem 5.1.** *The procedure for constructing $\mathcal{C}_k(t)$ provides anytime-valid coverage. That is, for any stopping time $T$:*

$$\mathbb{P}\left(\{l_j \mid r_j \leq k\} \subseteq \mathcal{C}_k(T)\right) \geq 1 - \alpha.$$

As additional data are collected, the rank confidence intervals $[L_j(t), U_j(t)]$ tighten, leading to a progressively smaller and more informative top-$k$ set. This formulation naturally motivates adaptive sampling: to further refine $\mathcal{C}_k(t)$, one should prioritize pairwise comparisons that are most likely to increase the lower rank bounds of models currently in $\mathcal{C}_k(t)$ or decrease the upper bounds of those outside it.

# 6. Simulation Results

Suppose there are $m$ models, each associated with an underlying latent score $\theta_j \in \mathbb{R}$. For $(j, k)$, the probability that item $j$ is preferred over item $k$ is modeled as $P(j \succ k) = \exp(\theta_j) / \{\exp(\theta_j) + \exp(\theta_k)\}$. To evaluate the effect of SERPANT, we report FWER and power, which is defined as the expectation of the proportion of correctly rejected partial order pairs. We repeat the simulation $N = 1000$ times.

We begin by evaluating the SERPANT algorithm using the *baseline e-process*. Additional results obtained with the covariate-assisted e-process, which further leverages side information to enhance power, are deferred to Appendix C. The latent scores are independently drawn from a normal distribution: $\theta_j \sim N(0, \sigma^2)$ for $j = 1, \ldots, m$. To examine the empirical behavior of the method, we conduct experiments with $m = 10$ models and vary the standard deviation $\sigma$ of the capability distribution over the set $\{0.1, 0.2, 0.5, 1.0\}$. Figure 1 illustrates the results under the baseline e-process scheme. In all figures, the shaded regions represent 95% confidence bands unless stated otherwise. The left panel confirms that the SERPANT algorithm successfully controls the FWER at the nominal level $\alpha = 0.1$ at every time point. The right panel displays the corresponding average power, which increases monotonically as more comparisons are collected. Larger values of $\sigma$ lead to faster power gains, while more homogeneous settings ($\sigma = 0.1$) require substantially more comparisons. Overall, these results confirm the theoretical guarantees of SERPANT and illustrate its effectiveness in distinguishing models with varying degrees of separability.

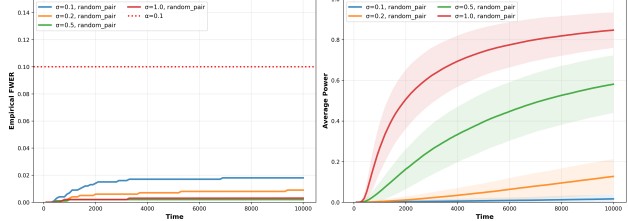

*Figure 1.* FWER and power over time under random-pair sampling e-processes.

We further investigate the SERPANT framework with the

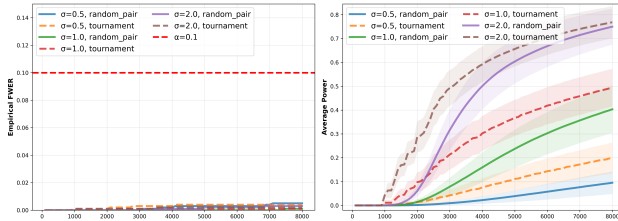

*Figure 2.* FWER and power over time: tournament vs. random-pair sampling e-processes.

tournament sampling strategy. We compare two sampling mechanisms: a baseline random pair selection and a tournament-style selection method. We choose $w_1 = 0.2$, $w_2 = 0.8$, $m = 20$ and $k = 0.9 \cdot |\mathcal{A}_t|$. The results, as summarized in figure 2, demonstrate that the tournament sampling consistently outperforms random pair selection in terms of achieved statistical power across all $\sigma$ settings.

Beyond full ranking, we evaluate the capability of the SER-PANT framework for Top-10 sets in a system with $m = 20$ models, comparing the baseline and tournament sampling strategies under varying levels of model heterogeneity. Figure 3 summarizes the Top-$k$ FWER (left panels) and the average size of the identified set (right panels) over time. The results demonstrate that the procedure successfully controls the error rate under the target level for both sampling methods across all settings. As expected, the tournament strategy produces smaller Top-10 sets than the random sampling method.

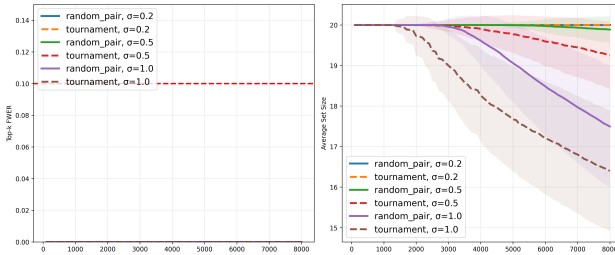

*Figure 3.* FWER and average Top-10 set size over time: tournament vs. random-pair sampling.

# 7. Real Experiment

To evaluate the effectiveness of the proposed SERPANT algorithm for ranking LLMs, we conduct real-data experiments on an open-domain question answering (QA) task using several widely adopted benchmarks. These datasets collectively provide a comprehensive assessment of model capabilities across different dimensions. Specifically, we employ: the `TriviaQA` benchmark (Joshi et al., 2017), which contains diverse fact-seeking questions suitable for evaluating factual accuracy; the `SQuAD2.0` dataset (Ra-

jpurkar et al., 2016; 2018) for testing reading comprehension; the `databricks-dolly-15k` dataset (Wang et al., 2023) to assess instruction-following and text generation; and the `MMLU` benchmark (Hendrycks et al., 2021) for measuring multi-disciplinary knowledge and reasoning.

We consider 10 open-source LLMs with varying architectures and parameter sizes, as shown in Table 1. For each question, models are compared in a pairwise manner based on their generated answers, and a preference label is obtained using `gpt-4o-mini` (OpenAI et al., 2024) as an automated judge. See Appendix C for query templates.

*Table 1.* Open-sourced LLMs evaluated in the experiments.

| Model | Reference |
|---|---|
| Qwen2.5-1.5B-Instruct | (Yang et al., 2024) |
| Qwen2.5-3B-Instruct | (Yang et al., 2024) |
| gemma-2-2b-it | (Team et al., 2024) |
| TinyLlama-1.1B-Chat-v1.0 | (Zhang et al., 2024) |
| gpt2 | (Radford et al., 2019) |
| DeepSeek-R1-Distill-Qwen-1.5B | (DeepSeek-AI et al., 2025) |
| Baguettotron | (PleIAs, 2024) |
| Llama-2-7b-hf | (Touvron et al., 2023) |
| SmolLM3-3B | (Bakouch et al., 2025) |
| glm-edge-4b-chat | (Du et al., 2022) |

*Table 2.* Partial-order recovery across four datasets at time $t$. Numbers denote the number of discovered pairwise orderings, with percentages relative to the total ordering.

| time ($t$) | TriviaQA | SQuAD2.0 | databricks | MMLU |
|---|---|---|---|---|
| 1000 | 18 (40%) | 18 (40%) | 28 (62%) | 27 (60%) |
| 2000 | 27 (60%) | 29 (64%) | 41 (91%) | 33 (73%) |
| 3000 | 29 (64%) | 37 (82%) | 43 (96%) | 35 (78%) |
| 4000 | 34 (76%) | 38 (84%) | 45 (100%) | 37 (82%) |
| 5000 | 36 (80%) | 38 (84%) | 45 (100%) | 41 (91%) |

Table 2 reports partial-order recovery statistics across four benchmark datasets under increasing comparison budgets $t$. For each budget $t$, we measure the number of pairwise relations in the latent model ranking that are correctly identified by the baseline e-process with random sampling. As the query budget increases, performance improves consistently across all datasets, demonstrating that additional comparisons yield more accurate partial-order estimation. Notably, Databricks and MMLU exhibit steeper gains at larger budgets, suggesting that richer instruction-following and multi-domain settings particularly benefit from increased comparison allocation.

After summarizing the total discoveries, we focus on two representative examples and report the remaining results in Appendix C. We first consider the `databricks-dolly-15k` dataset, which contains diverse prompts spanning conversational, reasoning, and task-oriented instructions. Using pairwise comparison feedback,

*Table 3.* Model ranking obtained from statistically significant pairwise comparisons on the `databricks-dolly-15k` dataset.

| Rank | Model Name |
|------|-----------|
| 1 | gemma-2-2b-it |
| 2 | Qwen2.5-3B-Instruct |
| 3 | glm-edge-4b-chat |
| 4 | Qwen2.5-1.5B-Instruct |
| 5 | TinyLlama-1.1B-Chat-v1.0 |
| 6 | SmolLM3-3B |
| 7 | Llama-2-7b-hf |
| 8 | DeepSeek-R1-Distill-Qwen-1.5B |
| 9 | Baguettotron |
| 10 | gpt2 |

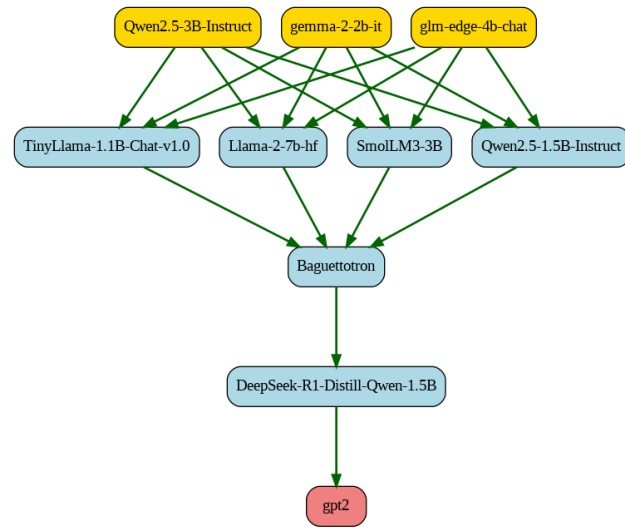

*Figure 4.* Directed acyclic graph (DAG) representing the inferred LLM ranking on a QA dataset at $t = 5000$. An edge $A \rightarrow B$ indicates that model $A$ significantly outperforms model $B$.

*Table 4.* Partial-order recovery across four datasets under random versus tournament sampling at time $t$. The numbers report discovered pairwise orderings (percentage of total).

| Dataset | Sampling | t=500 | t=1000 | t=1500 | t=2000 |
|---------|----------|-------|--------|--------|--------|
| Databricks | random | 5 (11%) | 28 (62%) | **38 (84%)** | **41 (91%)** |
| | tournament | **12 (27%)** | **32 (71%)** | 37 (82%) | **41 (91%)** |
| MMLU | random | 3 (7%) | **27 (60%)** | **31 (69%)** | **33 (73%)** |
| | tournament | **8 (18%)** | **27 (60%)** | **31 (69%)** | 32 (71%) |
| SQuAD2.0 | random | 4 (9%) | 18 (40%) | 23 (51%) | 29 (64%) |
| | tournament | **9 (20%)** | **21 (47%)** | **26 (58%)** | **30 (67%)** |
| TriviaQA | random | 4 (9%) | 17 (38%) | 18 (40%) | 23 (51%) |
| | tournament | **10 (22%)** | **23 (51%)** | **28 (62%)** | **32 (71%)** |

we evaluate competing models and apply our sequential e-process–based testing procedure to identify statistically significant performance differences. The resulting model ordering at $t = 5000$ is summarized in Table 3. The resulting ranking is consistent with model scale and instruction-tuning: larger and more recent instruction-tuned models (e.g., `gemma-2-2b-it`, `Qwen2.5-3B-Instruct`) appear at the top, while smaller or non–instruction-tuned baselines (e.g., `gpt2`) rank lower. Importantly, models with similar capacities (e.g., `Qwen2.5` variants) are ordered closely, indicating that the procedure avoids over-separating models when evidence is weak. This behavior aligns to control false discoveries while producing a meaningful and interpretable global ordering. Figure 4 shows the result after running the SERPANT algorithm over $t = 5000$ pairwise comparisons on the `TriviaQA` benchmark. At this stage, SERPANT identifies 36 statistically significant pairwise relations and outputs a directed acyclic graph (DAG) representing a partial order over models. In the DAG, a directed edge $A \rightarrow B$ indicates that model $A$ is inferred to significantly outperform model $B$ using anytime-valid testing. The resulting structure reveals a clear multi-layer hierarchy. Larger and instruction-tuned models, such as `Qwen2.5-3B-Instruct`, `gemma-2-2b-it`, and `glm-edge-4b-chat`, consistently dominate smaller or less specialized models and appear in the top layer. The partial order is interpretable, capturing clear dominance relationships where performance differences are big, while leaving incomparable pairs when evidence is insufficient.

**Comparison of Random and Tournament Sampling.** We further examine how the choice of sampling strategy affects the speed at which SERPANT resolves pairwise comparisons. Using four different datasets, we compare *random sampling* and *tournament sampling* by tracking the number of rejected pairwise null hypotheses over time. Specifically, at a sequence of pre-specified time points, we record the size of the selected partial order. This quantity reflects how quickly the algorithm accumulates statistically significant evidence and recovers the underlying ranking structure.

The results are summarized in Table 4. Across most time horizons, tournament sampling yields a larger recovered partial order than random sampling, indicating more efficient allocation of comparison queries toward informative model pairs. These findings corroborate the advantage of adaptive sampling strategies for accelerating inference, while preserving the anytime-valid error guarantees of SERPANT. In general, tournament sampling improves statistical power under limited comparison budgets, although, due to inherent sampling randomness, its advantage may be mild in some dataset-specific scenarios.

Overall, this experiment demonstrates that SERPANT can efficiently recover a meaningful and interpretable partial ranking among LLMs from a limited number of comparisons, highlighting its practical utility for real-world model evaluation scenarios where exhaustive comparisons are infeasible.

**Benefit of transitivity-based pruning.** We use Databricks to evaluate the power gain from transitivity-based pruning and to assess whether the transitivity assumption is empirically reasonable. The pruning step assumes transitivity to propagate discovered pairwise preferences and remove comparisons that are already implied by the current partial order. As shown in Table 5, disabling pruning leads to a substantial loss of power, demonstrating that transitivity-based pruning provides a significant efficiency gain. Moreover, we do not observe any contradictory pairwise relations, suggesting that transitivity is a reasonable assumption.

*Table 5.* Partial-order recovery across Databricks-dolly-15k dataset under random sampling at time $t$. The numbers report discovered pairwise orderings (percentage of total).

| Dataset | Transitivity | t=1000 | t=2000 | t=3000 |
|---|---|---|---|---|
| Databricks | Prune | **28 (62%)** | **41 (91%)** | **43 (96%)** |
| | Without prune | 24 (53%) | 35 (78%) | 38 (84%) |

**Public leaderboards.** To validate SERPANT in an LLM-as-a-judge evaluation setting, we use the public LMArena text-task leaderboard as the ground-truth model quality. We select 10 models from the leaderboard and compare their public rankings with the rankings recovered by SERPANT on the Databricks-Dolly-15k questions. The LMArena ranking as of March 25, 2026, is reported in Table 6.

*Table 6.* Ranking of 10 models selected from the Arena leaderboard for text tasks.

| Rank | Model Name |
|---|---|
| 1 | Llama-3.1-Tulu-3-8B |
| 2 | Granite-3.1-8B-Instruct |
| 3 | Gemma-2-2B-IT |
| 4 | Mistral-7B-Instruct-v0.2 |
| 5 | Qwen1.5-7B-Chat |
| 6 | Zephyr-7B-Beta |
| 7 | SmolLM2-1.7B-Instruct |
| 8 | Llama-2-7B-Chat |
| 9 | Qwen1.5-4B-Chat |
| 10 | FastChat-T5-3B-v1.0 |

We compare SERPANT with an empirical win-rate baseline that ranks models by their observed pairwise winning rates. The results in Table 7 show that SERPANT recovers a leaderboard-consistent ranking structure while maintaining rigorous anytime-valid error control. In contrast, the empirical baseline does not control the error rate and yields a non-negligible number of false discoveries during evaluation. The directed acyclic graph produced by SERPANT at time $t = 4000$ is shown in Figure 5. Overall, the recovered ranking is broadly consistent with the LMArena leaderboard, with only one discrepancy. This difference may be attributable to task-specific effects: our evaluation is restricted to Databricks, whereas LMArena aggregates preferences over a broader collection of tasks.

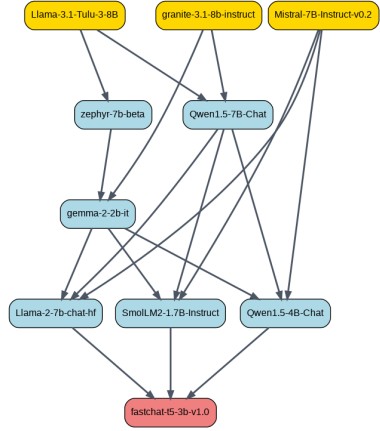

*Figure 5.* DAG of the ranking recovered for selected models on the Databricks-Dolly-15k dataset at $t = 4000$.

*Table 7.* Comparison of true and false discoveries made by SERPANT and the empirical win-rate baseline over time.

| Method | Metric | $t = 1000$ | $t = 2000$ | $t = 3000$ | $t = 4000$ |
|---|---|---|---|---|---|
| SERPANT | true | 16 (36%) | 29 (64%) | 32 (71%) | 32 (71%) |
| | false | 0 (0%) | 0 (0%) | 1 (2%) | 1 (2%) |
| Empirical | true | 39 (87%) | 40 (89%) | 40 (89%) | 41 (91%) |
| | false | 6 (13%) | 5 (11%) | 5 (11%) | 4 (9%) |

## 8. Discussion

Several open directions emerge from this work. First, while we focus on anytime-valid FWER control, large-scale evaluation platforms may benefit from weaker but more powerful error criteria, such as anytime-valid *false discovery rate control*. Developing such procedures while preserving time-uniform guarantees remains an open challenge.

Second, our framework focuses on inference for the overall average preference. In practice, human preferences are often context-dependent. Extending anytime-valid ranking inference to content-specific preferences constitutes an essential direction for future research.

Finally, LLM-as-a-judge evaluators are increasingly adopted to enable scalable preference collection. Incorporating corrections for judge-induced biases into anytime-valid inference pipelines is essential for building reliable online evaluation systems in practice.

## Impact Statement

This paper presents a methodological and theoretical study of ranking LLMs through online evaluation. The goal is to advance foundational understanding of model ranking and evaluation. Any empirical rankings are included solely for methodological illustration and are sensitive to modeling assumptions, sampling procedures, and evaluation random-

ness; they do not represent the authors' value judgments or constitute guidance for real-world deployment or policy decisions. The broader societal implications of this work are indirect and mediated through future applications by practitioners.

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

# A. Theoretical Proofs

This appendix provides detailed proofs for the theoretical results presented in the main paper. We begin by recalling the key definition: nonnegative supermartingale and e-processes.

**Definition A.1** (Nonnegative Martingale with respect to the Natural Filtration). Let $(\Omega, \mathcal{F}, \mathbb{P})$ be a probability space, and let $\{\mathcal{F}_t\}_{t\geq 0}$ be a filtration, i.e., an increasing sequence of $\sigma$-algebras $\mathcal{F}_0 \subseteq \mathcal{F}_1 \subseteq \cdots \subseteq \mathcal{F}_t$. A stochastic process $\{E_t\}_{t\geq 0}$ is called a *nonnegative martingale* with respect to the filtration $\{\mathcal{F}_t\}_{t\geq 0}$ if it satisfies:

1. (**Nonnegativity**) $E_t \geq 0$ almost surely for all $t \geq 0$.

2. (**Adaptedness**) For each $t \geq 0$, $E_t$ is $\mathcal{F}_t$-measurable.

3. (**Integrability**) $\mathbb{E}[E_t] < \infty$ for all $t \geq 0$.

4. (**Martingale property**)
$$\mathbb{E}[E_{t+1} \mid \mathcal{F}_t] = E_t \quad \text{almost surely, for all } t \geq 0.$$

**Definition A.2** (e-Process). Let $(\Omega, \mathcal{F}, (\mathcal{F}_t)_{t\in\mathcal{T}})$ be a filtered probability space, and let $\mathcal{P}$ be a family of probability measures. An adapted sequence of nonnegative random variables $E = (E_t)_{t\in\mathcal{T}}$ is called an *e-process* (with respect to $\mathcal{P}$) if it satisfies either of the following equivalent conditions:

1. For every stopping time $\tau$ taking values in $\mathcal{T}$ and for every $P \in \mathcal{P}$,
$$\mathbb{E}_P[E_\tau] \leq 1.$$

2. There exists a family of nonnegative martingales $\{M^P\}_{P\in\mathcal{P}}$ such that, for every $P \in \mathcal{P}$ and all $t \in \mathcal{T}$,
$$E_t \leq M_t^P \qquad P\text{-almost surely.}$$

Given the above definition, we first prove Proposition 3.1, which serves as the foundation for establishing anytime FWER control.

*Proof of Proposition 3.1.* We verify the four defining properties of a nonnegative e-Process:

**Nonnegativity and Adaptivity:** Since $e_{j,k}^t$ is defined as a ratio of nonnegative quantities, it is $\mathcal{F}_t$-measurable. By the definition,
$$e_{j,k}^t = \frac{\int_{0.5}^1 L(p; S_t, t)\, g(p)\, dp}{\sup_{p\in[0,0.5]} L(p; S_t, t)},$$

where both the numerator and denominator are strictly nonnegative. Hence $e_{j,k}^t$ is a nonnegative process adapted to the filtration $\mathcal{F}_t$.

**Integrability:** For all $t \geq 0$ and $p_0 \in [0, 0.5]$, the expectation $\mathbb{E}_{p_0}[e_{j,k}^t]$ is finite. This follows from the fact that both the integral in the numerator and the supremum in the denominator are bounded on their compact parameter domains.

**Martingale Property:** Let $L_t(p) = p^{S_t}(1-p)^{n_t - S_t}$ denote the likelihood function. For each $q \in [0, 0.5]$, we define
$$M_t(q) := \int_{0.5}^1 \frac{L_t(p)}{L_t(q)}\, g(p)\, dp$$

and thus
$$e_{j,k}^t := \inf_{q\in[0,0.5]} M_t(q) \leq M_t(p_0),$$

where $p_0$ is the true probability under the null. Before establishing that $\{e_{j,k}^t\}_{t\geq 0}$ is a nonnegative supermartingale under the composite null, we first show that $M_t(p_0)$ forms a martingale. Note that $L_t(p) = L_{t-1}(p) \cdot p^{z_{j,k}^t}(1-p)^{1-z_{j,k}^t}$. Therefore, we can write
$$M_t(p_0) = \frac{\int_{0.5}^1 L_{t-1}(p) p^{z_{j,k}^t}(1-p)^{1-z_{j,k}^t} g(p) dp}{L_{t-1}(p_0) p_0^{z_{j,k}^t}(1-p_0)^{1-z_{j,k}^t}}.$$

Taking the conditional expectation:

$$\mathbb{E}_{p_0}[M_t(p_0) \mid \mathcal{F}_{t-1}] = \mathbb{E}_{p_0}\left[\left.\frac{\int_{0.5}^1 L_{t-1}(p)p^{z_t}(1-p)^{1-z_t}g(p)dp}{L_{t-1}(p_0)p_0^{z_t}(1-p_0)^{1-z_t}} \right| \mathcal{F}_{t-1}\right].$$

It follows that

$$\mathbb{E}_{p_0}[M_t(p_0) \mid \mathcal{F}_{t-1}] = \frac{1}{L_{t-1}(p_0)} \int_{0.5}^1 L_{t-1}(p)g(p) \cdot \mathbb{E}_{p_0}\left[\left.\frac{p^{z_{j,k}^t}(1-p)^{1-z_{j,k}^t}}{p_0^{z_{j,k}^t}(1-p_0)^{1-z_{j,k}^t}} \right| \mathcal{F}_{t-1}\right] dp,$$

where we use the Fubini-Tonelli theorem (applicable due to nonnegativity) and exchange the order of integration and expectation. Directly computing the inner expectation, we have

$$\mathbb{E}_{p_0}\left[\frac{p^{z_{j,k}^t}(1-p)^{1-z_{j,k}^t}}{p_0^{z_{j,k}^t}(1-p_0)^{1-z_{j,k}^t}}\right] = p_0 \cdot \frac{p}{p_0} + (1-p_0) \cdot \frac{1-p}{1-p_0} = p + (1-p) = 1.$$

Thus, we have

$$\mathbb{E}_{p_0}[M_t(p_0) \mid \mathcal{F}_{t-1}] = \frac{1}{L_{t-1}(p_0)} \int_{0.5}^1 L_{t-1}(p)g(p)dp = M_{t-1}(p_0).$$

Thus, we know $M_t(p_0)$ forms a martingale. Furthermore,

$$\mathbb{E}_{p_0}[M_t(p_0)] = \int_{0.5}^1 g(p)dp = \int_{0.5}^1 2 \cdot dp = 1.$$

By definition, we know $\{e_{j,k}^t\}_{t\geq 0}$ is an e-process. $\qquad\square$

Then, we consider the covariate-assisted e-process setting, in which the weighting function is given by an adaptive (data-dependent) estimator. The following theorem establishes its theoretical validity.

**Theorem A.3** (Validity of the covariate-assisted adaptive e-process). *Let $\{(z^t, x^t)\}_{t\geq 1}$ be a sequential process on a filtered probability space $(\Omega, \mathcal{F}, \{\mathcal{F}_t\}_{t\geq 0}, \mathbb{P})$, where $(z^t, x^t)$ is $\mathcal{F}_t$-measurable and $z^t$ denotes the comparison outcome at time t, and we omit the model index subscripts for notational simplicity. For each $t \geq 1$, let $\hat{g}_t(\cdot)$ be an $\mathcal{F}_{t-1}$-measurable (possibly random) function constructed from the past data, then the e-process*

$$\tilde{e}_{j,i}^t = \frac{\prod_{s\in T_t} \int_{0.5}^1 p^{z_{j,k}^s}(1-p)^{1-z_{j,k}^s}\hat{g}_s(p)\,dp}{\sup_{p\in[0,0.5]} L(p; S_t, t)},$$

*with estimated weight function $\hat{g}_t(\cdot)$ is a valid e-process. The results in Theorem 3.2 and 5.1 hold for the covariate-assisted e-process.*

*Proof of Theorem A.3.* It remains to show that the covariate-assisted e-process using the estimated weight function $\hat{g}_t(\cdot)$ is a valid e-process. We verify the four defining properties of a nonnegative e-Process:

**Nonnegativity and Adaptivity:** Since $\tilde{e}_{j,i}^t$ is defined as a ratio of nonnegative quantities, it is $\mathcal{F}_t$-measurable. By the definition,

$$\tilde{e}_{j,i}^t = \frac{\prod_{s\in T_t} \int_{0.5}^1 p^{z_{j,k}^s}(1-p)^{1-z_{j,k}^s}\hat{g}_s(p)\,dp}{\sup_{p\in[0,0.5]} L(p; S_t, t)},$$

where both the numerator and denominator are strictly nonnegative. Hence $\tilde{e}_{j,i}^t$ is a nonnegative process adapted to the filtration $\mathcal{F}_t$.

**Integrability:** For all $t \geq 0$ and $p_0 \in [0, 0.5]$, the expectation $\mathbb{E}_{p_0}[\tilde{e}_{j,k}^t]$ is finite. This follows from the fact that both the integral in the numerator and the supremum in the denominator are bounded on their compact parameter domains.

**Martingale Property:** Let $L_t(p) = p^{S_t}(1-p)^{n_t-S_t}$ denote the likelihood function. For each $q \in [0, 0.5]$, we define

$$M_t(q) := \frac{\prod_{s=1}^t \int_{0.5}^1 p^{z_{j,k}^s}(1-p)^{1-z_{j,k}^s}\hat{g}_s(p)\,dp}{L(q; S_t, t)}$$

and thus

$$\tilde{e}_{j,k}^t := \inf_{q \in [0,0.5]} M_t(q) \leq M_t(p_0),$$

where $p_0$ is the true probability under the null. Before establishing that $\{\tilde{e}_{j,k}^t\}_{t \geq 0}$ is a nonnegative supermartingale under the composite null, we first show that $M_t(p_0)$ forms a martingale. Note that $L_t(p) = L_{t-1}(p) \cdot p^{z_{j,k}^t}(1-p)^{1-z_{j,k}^t}$. Therefore, we can write

$$M_t(p_0) = \frac{\prod_{s \in T_t} \int_{0.5}^1 p^{z_{j,k}^s}(1-p)^{1-z_{j,k}^s} \, \hat{g}_s(p) \, dp}{L_{t-1}(p_0) p_0^{z_{j,k}^t}(1-p_0)^{1-z_{j,k}^t}}.$$

Taking the conditional expectation:

$$\mathbb{E}_{p_0}[M_t(p_0) \mid \mathcal{F}_{t-1}] = \mathbb{E}_{p_0}\left[ \frac{\prod_{s \in T_t} \int_{0.5}^1 p^{z_{j,k}^s}(1-p)^{1-z_{j,k}^s} \, \hat{g}_s(p) \, dp}{L_{t-1}(p_0) p_0^{z_{j,k}^t}(1-p_0)^{1-z_{j,k}^t}} \,\middle|\, \mathcal{F}_{t-1} \right].$$

It follows that

$$\mathbb{E}_{p_0}[M_t(p_0) \mid \mathcal{F}_{t-1}] = \frac{\prod_{s \in T_t} \int_{0.5}^1 p^{z_{j,k}^s}(1-p)^{1-z_{j,k}^s} \, \hat{g}_s(p) \, dp}{L_{t-1}(p_0)} \int_{0.5}^1 \hat{g}_t(p) \cdot \mathbb{E}_{p_0}\left[ \frac{p^{z_{j,k}^t}(1-p)^{1-z_{j,k}^t}}{p_0^{z_{j,k}^t}(1-p_0)^{1-z_{j,k}^t}} \,\middle|\, \mathcal{F}_{t-1} \right] dp,$$

where we use the Fubini-Tonelli theorem (applicable due to nonnegativity) and exchange the order of integration and expectation. Directly computing the inner expectation, we have

$$\mathbb{E}_{p_0}\left[ \frac{p^{z_{j,k}^t}(1-p)^{1-z_{j,k}^t}}{p_0^{z_{j,k}^t}(1-p_0)^{1-z_{j,k}^t}} \right] = p_0 \cdot \frac{p}{p_0} + (1-p_0) \cdot \frac{1-p}{1-p_0} = p + (1-p) = 1.$$

Thus, we have

$$\mathbb{E}_{p_0}[M_t(p_0) \mid \mathcal{F}_{t-1}] = \frac{1}{L_{t-1}(p_0)} \int_{0.5}^1 L_{t-1}(p)\hat{g}_t(p) dp = M_{t-1}(p_0).$$

Thus, we know $M_t(p_0)$ forms a martingale. Furthermore,

$$\mathbb{E}_{p_0}[M_t(p_0)] = \int_{0.5}^1 g(p) dp = \int_{0.5}^1 2 \cdot dp = 1.$$

By definition, we know $\{\tilde{e}_{j,k}^t\}_{t \geq 0}$ is an e-process. $\qquad\square$

The following lemma establishes the most important property of an e-process.

**Lemma A.4** (Ville's inequality). *Let $\{M_t\}_{t \in \mathbb{N}}$ be an e-process for a model $\mathcal{P}$. Then, for every $\alpha \in (0, 1]$, the following three statements are equivalent:*

$$\sup_{P \in \mathcal{P}} P\left( \exists t \in \mathbb{N} : \, M_t \geq \frac{1}{\alpha} \right) \leq \alpha;$$

$$\sup_{P \in \mathcal{P}} P\left( \sup_{t \in \mathbb{N}} M_t \geq \frac{1}{\alpha} \right) \leq \alpha;$$

$$\sup_{P \in \mathcal{P}, \, \tau} P\left( M_\tau \geq \frac{1}{\alpha} \right) \leq \alpha,$$

*for every $\alpha \in (0, 1)$, where the supremum ranges over all $\mathcal{F}$-stopping times $\tau$ taking values in $\mathbb{N} \cup \{0\}$.*

We are now in a position to establish the main theorem.

*Proof of Theorem 3.2.* By definition, the FWER at time $t$ is the probability of making at least one Type I error, that is,

$$\text{FWER}(t) := \mathbb{P}\Big(\exists\,(j,k) \in \mathcal{H}_0^\star \text{ such that } H_0^{(j,k)} \text{ is rejected by time } t\Big).$$

A Type I error for hypothesis $H_0^{(j,k)}$ occurs when the procedure rejects $H_0^{(j,k)}$ despite it being true.

Let $T$ be an arbitrary (possibly random) stopping time, and let $\mathcal{I}_0$ denote the index set of true null hypotheses. A hypothesis may be rejected either through a *direct rejection* (when an e-process crosses the threshold $K$) or a *consequential rejection* arising from logical propagation. An error from a consequential rejection of $H_0^{(j,k)}$ constitutes an error only if at least one of its antecedent hypotheses (e.g., $H_0^{(j,k)}$) was wrongly rejected. If all antecedents are false nulls, the propagated rejection cannot yield an error. Therefore, any Type I error—whether direct or consequential—necessarily implies that at least one direct rejection of a true null hypothesis has occurred.

Let $\mathcal{E}$ denote the event that at least one Type I error is made by time $T$, and let $\mathcal{E}_{\text{direct}}$ denote the event that at least one true null hypothesis in $\mathcal{I}_0$ is rejected directly via its e-process crossing the threshold $K$. As argued above, any Type I error must originate from a direct rejection of a true null; hence $P(\mathcal{E}) = P(\mathcal{E}_{\text{direct}})$. We can bound the probability of $\mathcal{E}_{\text{direct}}$ using the union bound:

$$P(\mathcal{E}_{\text{direct}}) = \mathbb{P}\left(\bigcup_{(j,k)\in\mathcal{I}_0} \left\{\sup_{t\leq T} e_{j,k}^t \geq K\right\}\right) \leq \sum_{(j,k)\in\mathcal{I}_0} \mathbb{P}\left(\sup_{t\leq T} e_{j,k}^t \geq K\right).$$

For any $(j,k) \in \mathcal{I}_0$, the process $(e_{j,k}^t)_{t\geq 0}$ is a non-negative test martingale. By Ville's inequality (Lemma A.4), for any stopping time $T$ and any constant $K > 0$:

$$\mathbb{P}\left(\sup_{t\leq T} e_{j,k}^t \geq K\right) \leq \frac{1}{K}.$$

The number of true null hypotheses, $|\mathcal{I}_0|$, is bounded by the total number of hypotheses, $m(m-1)$. Applying this bound to our sum:

$$\sum_{(j,k)\in\mathcal{I}_0} \mathbb{P}\left(\sup_{t\leq T} e_{j,k}^t \geq K\right) \leq \sum_{(j,k)\in\mathcal{I}_0} \frac{1}{K} = \frac{|\mathcal{I}_0|}{K} \leq \frac{m(m-1)}{K}.$$

By our choice of the rejection threshold $K = \frac{m(m-1)}{\alpha}$, we have:

$$\text{FWER}(T) = P(\mathcal{E}) \leq \frac{m(m-1)}{m(m-1)/\alpha} = \alpha.$$

This holds for any stopping time $T$, establishing anytime-valid control of the FWER. $\qquad\square$

In the following, we prove the corollary by reinterpreting the ranking error event in (5) within our framework of pairwise hypothesis testing.

*Proof of Corollary 3.3.* We reinterpret the events in (5) within the pairwise hypothesis testing framework introduced earlier.

- The event "$l_i \prec l_j$" indicates that $l_j$ is not strictly preferred to $l_i$. Under the Bernoulli preference model, this corresponds to $p_{j,i} \leq 0.5$, which is precisely the condition under which the null hypothesis $H_0^{(j,i)}$ is *true*.

- The event that the procedure concludes $l_j \succeq l_i$ means that a preference for $l_j$ over $l_i$ has been established by the procedure. By construction, this occurs if and only if the null hypothesis $H_0^{(j,i)}$ is *rejected*.

Consequently, the compound event "$l_i \prec l_j$ but the procedure concludes $l_j \succeq l_i$" is equivalent to the "event that $H_0^{(j,i)}$ is true and is nonetheless rejected, "which constitutes a Type I error for the hypothesis $H_0^{(j,i)}$.

The statement in (5), which begins with $\exists\,i \neq j$, asks for the probability that at least one such error occurs across all ordered pairs. By definition, this probability is the FWER.

By Theorem 3.2, the proposed procedure controls the FWER at level $\alpha$ uniformly over all stopping times. The corollary, therefore, follows immediately. $\qquad\square$

In the following, we prove that the proposed top-$k$ selection procedure controls FWER at level $\alpha$.

*Proof of Theorem 5.1.* We establish the error control guarantee for the proposed top-$k$ selection procedure. Let $\mathcal{E}$ denote the event that at least one true top-$k$ model is excluded from the confidence set $\mathcal{C}_k(T)$. That is, there exists an index $j^*$ such that the true rank satisfies $r_{j^*} \leq k$, but $l_{j^*} \notin \mathcal{C}_k(T)$.

By the definition of $\mathcal{C}_k(T)$, the condition $l_{j^*} \notin \mathcal{C}_k(T)$ implies that the lower confidence bound on its rank exceeds $k$, namely $L_{j^*}(T) > k$. Since $L_{j^*}(T) = 1 + N_{j^*}^+(T)$, this entails that by time $T$ the procedure has rejected at least $k$ hypotheses of the form $H_0^{(i,j^*)}$, concluding that at least $k$ other models are strictly preferred to $l_{j^*}$. If all such rejections were correct, then the true rank of $l_{j^*}$ would be at least $k + 1$, contradicting the assumption that $r_{j^*} \leq k$. Hence, the event $\mathcal{E}$ can occur only if at least one of the rejections contributing to $N_{j^*}^+(T)$ is a Type I error.

By Theorem 3.2, the probability that at least one Type I error occurs anywhere in the system by time $T$ is bounded above by $\alpha$. Since $\mathcal{E}$ implies the occurrence of at least one Type I error, it follows that $\mathbb{P}(\mathcal{E}) \leq \alpha$.

Consequently, the complementary event—that all true top-$k$ models are contained in $\mathcal{C}_k(T)$—satisfies

$$\mathbb{P}(\{l_j \mid r_j \leq k\} \subseteq \mathcal{C}_k(T)) = 1 - \mathbb{P}(\mathcal{E}) \geq 1 - \alpha,$$

uniformly over all stopping times $T$. $\qquad\square$

# B. Experimental Details

## B.1. Tournament sampling for SERPANT

Algorithm 2 describes a tournament-style sampling scheme that adaptively allocates comparison queries across active pairs as discussed in the main document. The scheme alternates between exploration rounds, in which pairs are sampled uniformly from the active set, and exploitation rounds, in which pairs are prioritized according to a composite score designed to target informative comparisons. Specifically, the base cap $B$ is the maximum number of comparison samples that may be allocated to a selected pair $(j, k)$ during a single exploration or exploitation selection. For exploration rounds, we set the maximum samples to the base cap $B$, while for exploitation rounds, we dynamically set the maximum samples $B_s$ according to signal strength $S^t(j, k)$. The rule choosing $B_s$ is empirical; if $S(j, k) \geq 0.6$, we set the maximum samples to $\infty$. If $0.4 \leq S(j, k) < 0.6$, we set $B_s$ to $3B$. And if $S(j, k) < 0.4$ then $B_s = B$. Sampling cap controls the granularity of sampling and prevents any single pair from consuming an excessive number of queries. The sampling proceeds in rounds, where each round corresponds to a complete pass through the current set of active pairs, and a bracket $\mathcal{B}$ maintains a queue of pairs selected for comparison in the current round. The exploration frequency is controlled by the exploration interval $F$, which dictates that the algorithm enters an exploration phase every $F$ rounds. Importantly, because SERPANT relies on e-processes, adaptive sampling and optional stopping do not affect the anytime FWER guarantee; tournament sampling is introduced solely to improve sample efficiency.

## B.2. Evaluation metrics and presentation of results

**Numerical evaluation.** For the simulation study, we evaluate the performance of SERPANT using power and family-wise error rate (FWER), both assessed in an anytime fashion. Let $\mathcal{R}_t$ denote the set of rejected hypotheses by time $t$, which is introduced in the Algorithm 1 and Algorithm 2. The power at time $t$ is defined as

$$\text{Power}_t = \mathbb{E}\left[ \frac{|\{(j, k) : \widehat{C}_{j,k}^t = C_{j,k} \text{ and } j \neq k\}|}{M(M-1)/2} \right].$$

To approximate this quantity empirically, we repeat each simulation $N = 1000$ times and report the Monte Carlo average. The estimated power at time $t$ is given by

$$\widehat{\text{Power}}_t = \frac{1}{N} \sum_{N=1} \frac{|\{(j, k) : \widehat{C}_{j,k}^t = C_{j,k} \text{ and } j \neq k\}|}{M(M-1)/2}.$$

---

**Algorithm 2** Tournament Sampling for SERPANT

---

**Require:** Models $\{l_1, \ldots, l_m\}$; FWER level $\alpha \in (0, 1)$; base cap $B$; exploration interval $F$; weights $w_1, w_2$; Selection number $q$; Exploration quantile parameter $l$.

1: Initialize $t \leftarrow 0$, round $\leftarrow 0$, $K \leftarrow \frac{m(m-1)}{\alpha}$, $e^0_{j,k} \leftarrow 1$, $\widehat{C}^0_{j,k} \leftarrow 0$, $s_{j,k} \leftarrow 0$, $n_{j,k} \leftarrow 0$ for all $j \neq i$, and bracket $\mathcal{B} \leftarrow \emptyset$.

2: **while** data arrive and $\widehat{C}^t_{j,k} = 0$ for some $j \neq k$: **do**

3:     $t \leftarrow t + 1$.

4:     **if** $\mathcal{B} = \emptyset$ **then**

5:         round $\leftarrow$ round $+1$, $n_{j,k} \leftarrow 0$ for $j \neq k$.

6:     **end if**

7:     **if** round $\bmod F = 1$ **then**

8:         select the bottom $1 - l$ proportion of pairs with the smallest sample counts from $\{(j, k) : \widehat{C}^t_{j,k} = 0 \text{ and } j \neq k\}$ and assign cap $B$ to each selected pair.

9:     **else**

10:         **for** each $(j, k) \in \{(j, k) : \widehat{C}^t_{j,k} = 0 \text{ and } j \neq k\}$ **do**

11:             $Q^t(j, k) \leftarrow w_1 U^t(j, k) + w_2 S^t(j, k)$.

12:         **end for**

13:     **end if**

14:     Select top-$q$ pairs by $\{Q^t(j, k)\}$ and dynamically assign cap $B_s$ to each selected pair according to $S^t(j, k)$.

15:     **for** each selected $(j, k)$ **do**

16:         Append $(j, k)$ to $\mathcal{B}$

17:     **end for**

18:     Take first pair $(j, k)$ from $\mathcal{B}$ and remove it.

19:     **if** $\widehat{C}^t_{j,k} = 0$ and $n^t_{j,k} \leq B$ **then**

20:         Sample $z^t_{j,k}$ and $n_{j,k} \leftarrow n_{j,k} + 1$.

21:     **end if**

22:     Update counters $n^t_{j,k}$ and e-values $e^t_{j,k}$.

23:     **if** $e^t_{j,k} \geq K$ **then**

24:         Reject $H_0^{(j,k)}$ and update $\widehat{C}^t_{j,k} = -1$, $\widehat{C}^t_{k,j} = 1$.

25:     **else if** $e^t_{k,j} \geq K$ **then**

26:         Reject $H_0^{(k,j)}$ and update $\widehat{C}^t_{j,k} = 1$, $\widehat{C}^t_{k,j} = -1$.

27:     **end if**

28:     Prune by transitivity: for any $h, l, m$, if $\widehat{C}^t_{h,l} = 1$ and $\widehat{C}^t_{l,m} = 1$, then set $\widehat{C}^t_{h,m} = 1$ and $\widehat{C}^t_{m,h} = -1$.

29: **end while**

**Ensure:** Estimated comparison matrix $\widehat{C}^t$.

---

We also evaluate error control using the family-wise error rate (FWER), defined as the probability of making at least one false rejection by time $t$. The empirical FWER is estimated as

$$\widehat{\text{FWER}}_t = \frac{1}{N} \sum_{N=1} \mathbb{I}\{\exists (k, k) : \widehat{C}^t_{j,k} \neq C_{j,k} \text{ and } j \neq k\}.$$

Together, these metrics quantify the trade-off between discovery power and error control over time. In particular, reporting power and FWER as functions of $t$ allows us to assess the anytime behavior of SERPANT under adaptive sampling.

**Graphical presentation of results.** We use the Directed Acyclic Graph (DAG) to represent rankings in the LLM evaluation. Let

$$V = \{l_1, \ldots, l_M\}$$

denote the set of models. At time $t$, we define the preference graph

$$G_t = (V, E_t),$$

where the edge set $E_t \subset V \times V$ is given by

$$E_t = \{(l_j, l_k) : (j, k) \in \mathcal{R}_t\}.$$

An edge $l_j \to l_k$ indicates that the null hypothesis $H_0^{(j,k)}$ has been rejected by time $t$, providing statistical evidence that $l_j \succ l_k$. By construction, $G_t$ represents the set of pairwise preferences inferred up to time $t$. Under the transitivity of preferences, the graph $G_t$ is a directed acyclic graph. To incorporate transitive implications, we maintain the transitive closure of $G_t$. In particular, if there exists a directed path from $l_k$ to $l_j$ in $G_t$, then the preference $l_k \succ l_j$ is implied and the corresponding ordered pair is removed from the active hypothesis set. This ensures that $G_t$ encodes a coherent partial order over the model set. The absence of a path between two vertices reflects insufficient evidence to determine their relative ordering.

**Topological order and ranking.** Given the DAG $G_t = (V, E_t)$, a *topological order* is any permutation $\sigma$ of $\{1, \ldots, M\}$ such that

$$(l_j, l_k) \in E_t \implies \sigma(j) < \sigma(k),$$

i.e., every directed edge points from an earlier to a later vertex in the ordering. When $G_t$ is acyclic, at least one topological order exists. Any such order is consistent with the currently established preferences and can be interpreted as a *ranking consistent with $G_t$*: models appearing earlier in $\sigma$ are ranked higher than those appearing later. If $G_t$ is not a total order, multiple topological orders may exist, reflecting unresolved comparisons. A topological order of the preference DAG $G_t = (V, E_t)$ admits a natural interpretation in terms of ranking tiers. Specifically, we define a sequence of disjoint subsets $\{\mathcal{T}_1, \mathcal{T}_2, \ldots\}$, referred to as *tiers*, constructed recursively as follows.

Tier $\mathcal{T}_1$ consists of all vertices in $G_t$ with in-degree zero. Models in $\mathcal{T}_1$ are not worse than any other model under the inferred preferences and are therefore ranked highest. After removing $\mathcal{T}_1$ and all outgoing edges from these vertices, tier $\mathcal{T}_2$ is defined as the set of vertices with in-degree zero in the remaining graph. Models in $\mathcal{T}_2$ are worse than those in $\mathcal{T}_1$ but not worse than any remaining models. This procedure is repeated until all vertices are assigned to a tier. Equivalently, the tiers correspond to layers of the DAG under successive elimination of minimal elements. Any topological order of $G_t$ is consistent with this tier structure: models in lower-indexed tiers always precede those in higher-indexed tiers, while the relative ordering of models within the same tier is not identified by the data. Thus, the tier representation provides a canonical summary of the partial ranking encoded by $G_t$, with each tier representing a set of models that are statistically indistinguishable in rank given the current evidence.

### B.3. Transitivity, pruning, and validity.

We clarify that the statistical validity of SERPANT does not rely on the transitivity assumption. The core inferential component of our method is the collection of sequential pairwise tests based on e-processes. By thresholding these e-processes with the Bonferroni-type level $K = m(m-1)/\alpha$, the procedure controls the family-wise error rate (FWER) uniformly over time, regardless of the order in which pairs are sampled and regardless of the stopping rule. This guarantee is purely pairwise and therefore remains valid even when the underlying preference relation is not transitive.

Transitivity is used only as an efficiency-enhancing device through the pruning step. When the preference relation is transitive, pairwise discoveries can be propagated: for example, once the procedure has established $l_a \succ l_b$ and $l_b \succ l_c$, transitivity implies $l_a \succ l_c$, so the corresponding comparison need not be sampled further. This pruning step reduces the number of active hypotheses and accelerates the construction of a coherent partial order, and eventually a linear ranking when enough relations are resolved. Such an assumption is natural in many objective evaluation domains, such as mathematical reasoning, coding, or factual question answering, where model quality is expected to induce an approximately transitive ordering.

However, transitivity need not hold in all evaluation settings, especially for subjective or preference-sensitive tasks. In such cases, the pruning step can simply be removed. The resulting unpruned version of SERPANT continues to perform anytime-valid sequential tests for all pairwise relations and still controls the FWER for the discovered directed preference graph. Consequently, if the underlying oracle exhibits cyclic preferences, such as $l_a \succ l_b$, $l_b \succ l_c$, and $l_c \succ l_a$, the unpruned procedure does not force these relations into a linear order; instead, it reports the statistically supported directed graph of pairwise preferences. Thus, transitivity affects only the efficiency and the interpretability of the output as a linear ranking, not the validity of the inferential guarantee.

This distinction also addresses potential path-dependence concerns. When transitivity is a valid structural assumption, pruning only propagates pairwise discoveries that are already supported by anytime-valid tests, so the final partial order is stable up to the small error probability controlled by the FWER. When transitivity is not appropriate, applying pruning may indeed introduce path dependence, since early discoveries could determine which later comparisons are skipped. For this

reason, we recommend using the unpruned version in domains where cyclic or context-dependent preferences are plausible. In summary, SERPANT can be viewed as a statistically valid directed-graph inference procedure without transitivity, and as an efficient ranking procedure with pruning when transitivity is a reasonable domain assumption.

### B.4. Marginalized rankings

LLM rankings are inherently task- and population-dependent. Accordingly, our ranking target is defined as the average pairwise winning probability, marginalized over the relevant user-prompt distribution. This marginalization is not a limitation; rather, it specifies the population-level target of evaluation. Although each user and prompt may induce a distinct subjective ranking, such instance-level rankings are typically neither broadly informative nor estimable from available data. In contrast, a marginalized ranking averages over the population of users, tasks, and prompts of interest, making it both practically meaningful and statistically estimable. For example, when ranking models for academic writing assistance, future users and prompts are generally unknown, so the relevant target is the average preference under the intended user-prompt distribution.

## C. Additional Numerical Results

### C.1. SERPANT with more models

We investigate the performance of the SERPANT algorithm across different numbers of models. The setting is discussed in the main document. We choose $M = 30$ and $M = 50$. The results are shown in Figure 6 and 7, respectively. As the number of models increases, it takes longer to compare them. However, these results confirm the theoretical guarantees of SERPANT and illustrate its effectiveness in distinguishing models with varying degrees of separability over time.

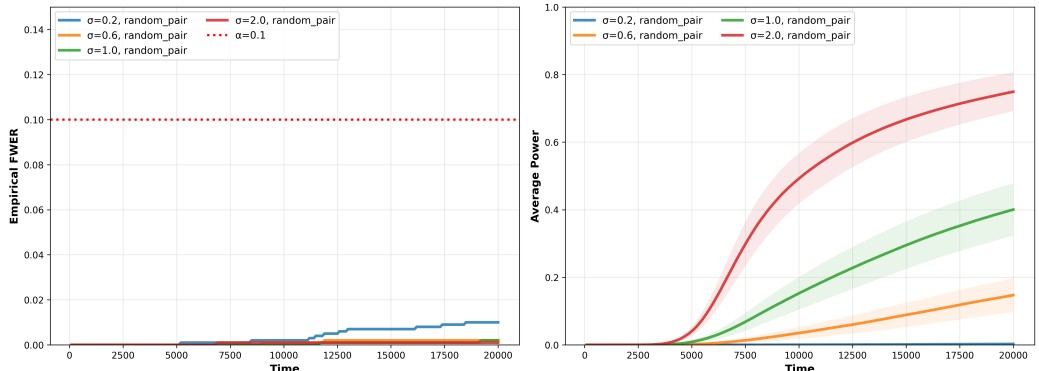

*Figure 6.* FWER and power curves of random-pair sampling over time when M=30.

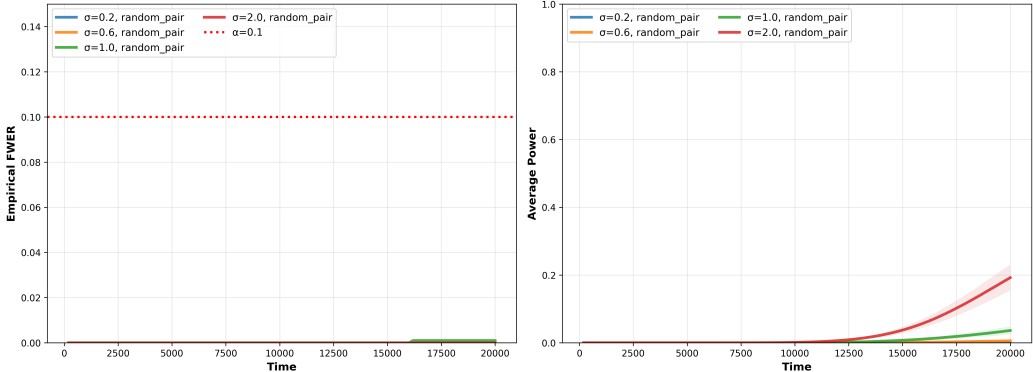

*Figure 7.* FWER and power curves of random-pair sampling over time when M=50.

## C.2. SERPANT with different weights

We further investigate how the weighting of exploration and exploitation in the tournament sampling scheme affects statistical power. In this experiment, we consider a family of priority rules of the form

$$\mathcal{Q}^t(j,k) = w_1 U^t(j,k) + w_2 S^t(j,k), \qquad w_1 + w_2 = 1,$$

which interpolates between pure uncertainty-driven sampling and signal-driven sampling.

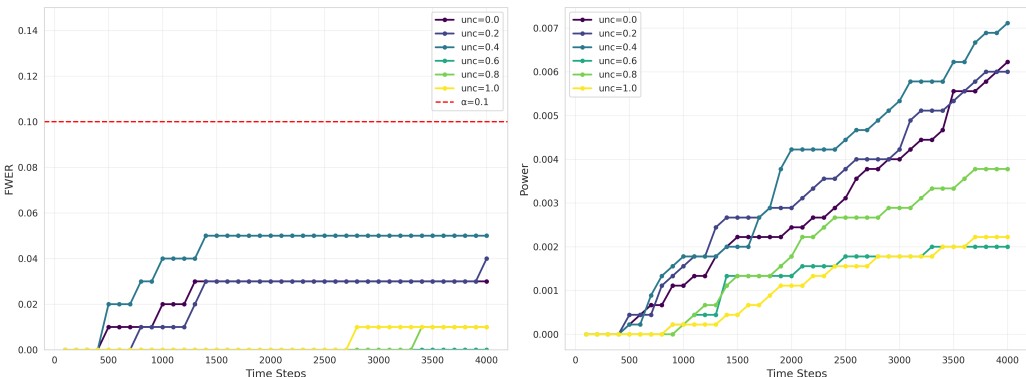

*Figure 8.* FWER and power curves of tournament sampling method over time under different weights when M=10 and $\sigma = 0.1$.

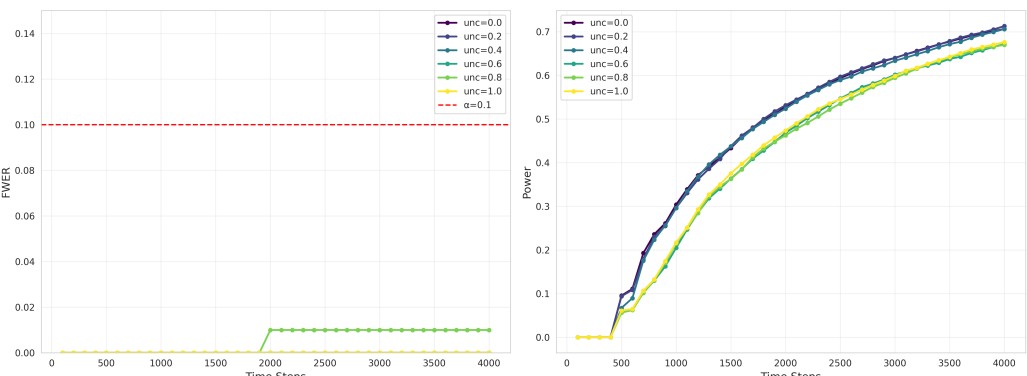

*Figure 9.* FWER and power curves of tournament sampling method over time under different weights when M=10 and $\sigma = 1$.

We evaluate this trade-off in a controlled simulation with $(M = 10)$ models under two noise regimes, corresponding to weak signal $(\sigma = 0.1)$ and strong signal $(\sigma = 1)$. As shown in Figure 8 and Figure 9, all choices of $(w_1, w_2)$ maintain valid FWER control, while their power trajectories differ substantially. When the signal is weak, allocating a moderate weight to uncertainty around $(w_1 = 0.4)$ yields the highest power, as it promotes exploration of ambiguous pairs that are most informative early on. In contrast, under strong signal conditions, prioritizing signal-dominated pairs $(w_1 \approx 0)$ is more effective, since the algorithm can rapidly exploit clear performance gaps. Overall, intermediate choices such as $(w_1 = 0.2)$ provide a robust compromise across regimes, delivering consistently strong power without sacrificing error control.

## C.3. SERPANT with covariates

We further investigate the performance of the SERPANT framework in a more realistic setting where model capabilities are influenced by observable covariates. In this experiment, we generate the true probabilities using a linear model structure: $Z_i = X_i \beta_i + \alpha_i$, where $X_i \sim N(0, \sigma_X)$, $\beta_i \sim N(0, \sigma_\beta)$, and $\alpha_i \sim N(0, \sigma_\alpha)$. This setup allows us to examine how the SERPANT algorithm performs when model strengths exhibit structured variation explained by external factors.

We evaluate the algorithm under different levels of covariate variability, setting $\sigma_X$ to 0.5, 1.0, and 2.0 while keeping $\sigma_\beta = 0.1$ and $\sigma_\alpha = 1$ fixed. As shown in Figure 10 and Figure 11, the SERPANT procedure maintains strict FWER control

at all time points for both sampling strategies—random pair selection and tournament selection—across all tested values of $\sigma_X$. This demonstrates the robustness of the e-value based methodology even when the underlying model capabilities are confounded by covariates. Regarding statistical power, the tournament sampling strategy consistently outperforms random pair selection. The advantage of tournament sampling becomes more pronounced as the covariate variability increases (e.g., $\sigma_X = 2.0$), suggesting that the adaptive sampling mechanism effectively utilizes the additional structure introduced by covariates to accelerate the discovery of true performance differences. Even when $\sigma_X = 0.5$, tournament sampling provides a noticeable improvement in power over the baseline random approach.

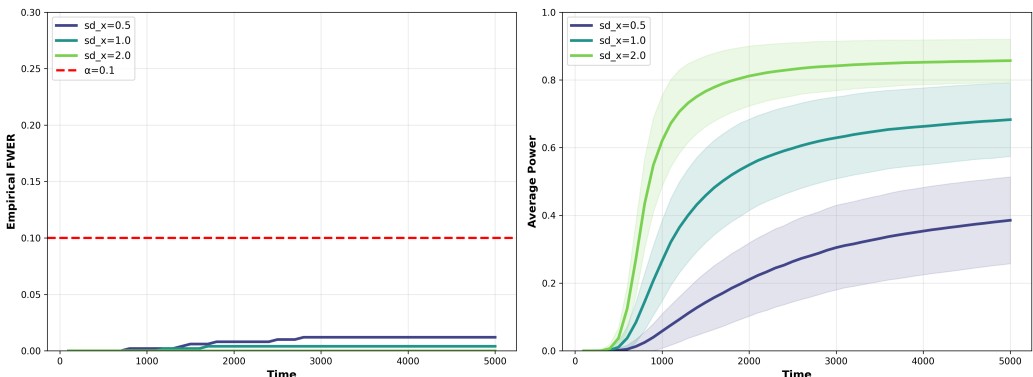

*Figure 10.* FWER and power curves of covariate method over time under different random X when M=10.

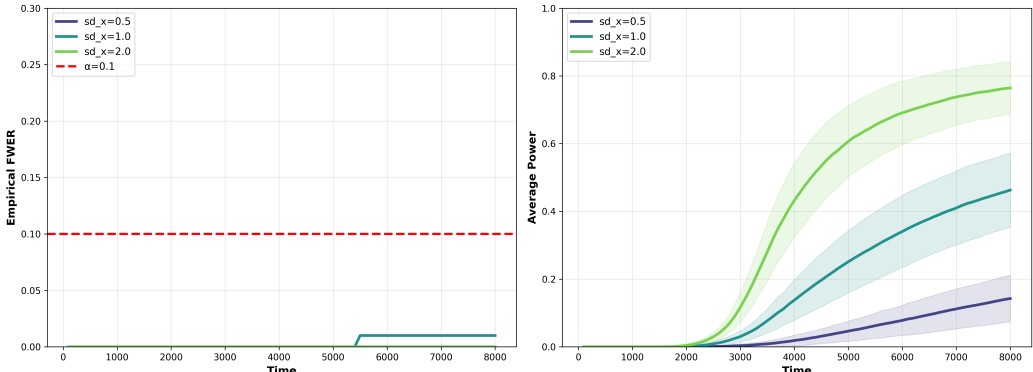

*Figure 11.* FWER and power curves of covariate method over time under different random X when M=20.

Figure 12 visualizes the empirical FWER (top row) and average power (bottom row) over time under the three covariate variability settings. Each column corresponds to a different value of $\sigma_X$, and within each setting we compare the original SERPANT procedure (RP-Orig), its covariate-aware variant (RP-Cov), as well as their counterparts under tournament sampling (Tour-Orig and Tour-Cov). The dashed red line marks the nominal FWER level $\alpha = 0.1$. Across all configurations, the empirical FWER curves remain well below the target level throughout the sampling process, confirming that anytime FWER control is preserved even in the presence of structured covariate effects. In contrast, the power curves show clear separation between sampling strategies: both covariate-aware methods achieve faster power accumulation than their original counterparts, and tournament sampling further amplifies this gain by prioritizing more informative comparisons. The magnitude of this benefit increases with smaller $\sigma_X$, indicating that SERPANT can exploit covariate-induced heterogeneity to accelerate correct discoveries in weak-signal settings while maintaining rigorous error control.

## C.4. Additional real data results

We further evaluate the SERPANT algorithm on additional real-world language modeling tasks to examine whether its ranking behavior generalizes across heterogeneous natural language generation settings. In the first experiment, we apply SERPANT to a reading-comprehension style QA task using the `SQuAD 2.0` dataset (Rajpurkar et al., 2016; 2018), which contains fact-seeking questions together with unanswerable cases and therefore imposes greater ambiguity and reasoning

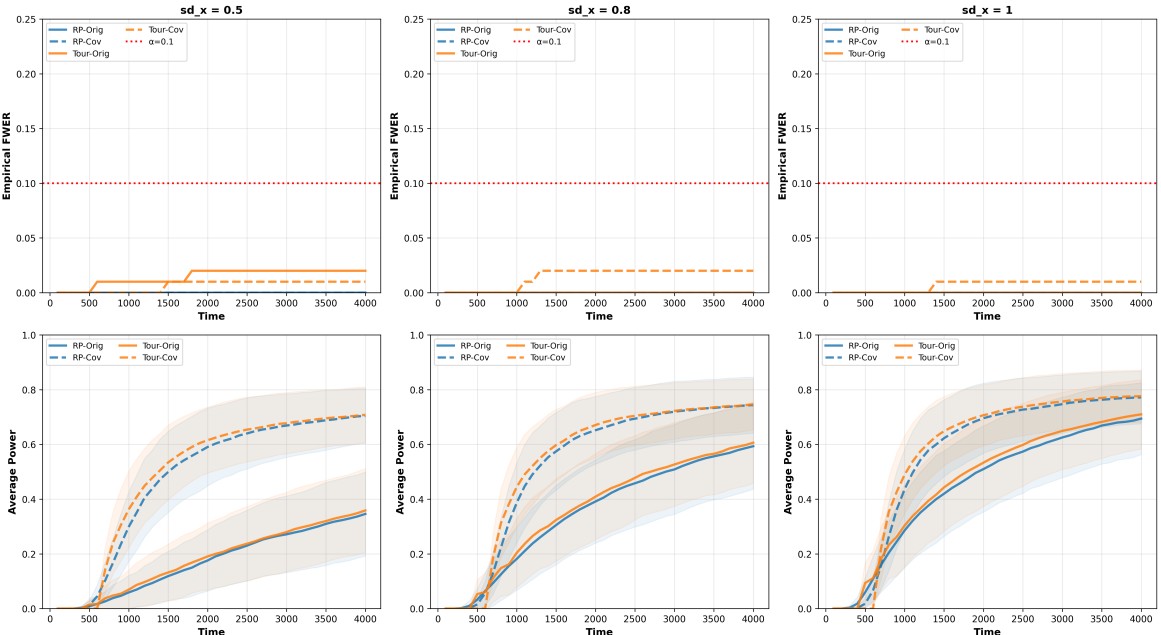

*Figure 12.* FWER and power curves of baseline vs. covariate method over time under different random X.

difficulty than fully-answerable benchmarks. Using the same pool of ten open-sourced LLMs and the same pairwise comparison protocol, SERPANT constructs the partial-order ranking shown in the first diagram at (t = 5000). The resulting hierarchy, as shown in Figure 13, exhibits a clear separation between higher- and lower-performing models, with statistically significant dominance relations established through the sequential e-value testing procedure.

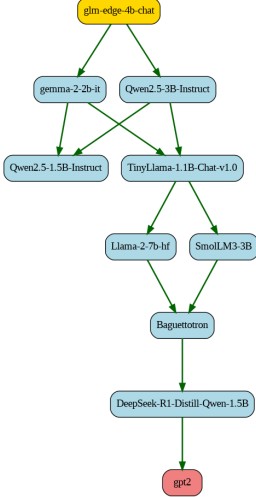

*Figure 13.* DAG of LLMs ranking over the SQuAD 2.0 dataset when $t = 5000$. Arrow $A \rightarrow B$ indicates model A outperforms model B.

In the second experiment, we consider an instruction-following generation task based on the `databricks-dolly-15k` dataset, which consists of diverse user prompts spanning conversational, reasoning, and task-oriented instructions. Evaluating the same model pool under pairwise comparison feedback, SERPANT produces the partial-order structure shown in the second diagram at (t = 5000). The results are shown in Figure 14. Compared with the QA setting, the induced ordering reveals slightly different dominance relationships, reflecting shifts in relative model strengths when assessed on instruction-style generation. Taken together, these two real-data studies demonstrate that SERPANT is able to recover stable and interpretable model ranking structures across distinct task domains while relying solely on noisy pairwise preference feedback.

In the third experiment, we evaluate the models on a multiple-choice question answering task using the MMLU benchmark (Hendrycks et al., 2021), which spans a broad collection of academic and professional domains. Under the same pairwise comparison protocol, SERPANT constructs the partial-order structure shown in Figure 15 at ($t = 5000$). Relative to the open-ended QA and instruction-following settings, the induced hierarchy on MMLU exhibits slight re-orderings among mid-tier models, suggesting that domain-knowledge and discrete option selection place different demands on model reasoning. Therefore, it demonstrates that SERPANT can recover stable model ranking structures across heterogeneous task formats using only noisy pairwise preference feedback.

### C.5. Query template and sample responses

To ensure consistent and fair evaluation across heterogeneous tasks, we adopt a unified query template for pairwise model comparison under human or model-based judgment. Each query presents a single prompt or question, along with two candidate responses generated by different models, and instructs the evaluator to select the better response based on task-specific criteria such as factual correctness, instruction following, reasoning quality, and clarity.

The templates are instantiated differently depending on the task type. For open-domain question-answering tasks (e.g., TriviaQA), the emphasis is on factual accuracy and completeness of the generated answer. For reading comprehension benchmarks (e.g., SQuAD 2.0), evaluators are instructed to assess relevance and correctness with respect to the given question and context. For multiple-choice reasoning datasets (e.g., MMLU), the primary criterion is whether the selected option is correct, with additional consideration of reasoning or explanation quality when provided. Finally, for instruction-following and text generation tasks (e.g., databricks-dolly-15k), evaluation focuses on how well the response adheres to the user instruction, its coherence and helpfulness, and overall language quality, reflecting the open-ended and generation-oriented nature of the task.

*Table 8.* Query template for the open-domain question answering task (TriviaQA).

```
You are an expert evaluator for open-domain question answering.  Compare two
responses (A and B) to the same question.  Which response is better overall based
on:
1.  Factual Correctness:  Is the answer factually correct?
2.  Completeness:  Does it sufficiently answer the question?
3.  Clarity:  Is the response clear and well-structured?
You MUST output only a single character:  'A' or 'B'.

Query:  Who was the last man to walk on the Moon?

Response A: <response_a>
Response B: <response_b>

Comparison:  <one-sentence comparison >
Preferred:  "A" or "B"
```

*Table 9.* Sample Responses of the open-domain question answering task (TriviaQA) Prompt and Pairwise Judgment

| Sample Responses of a TriviaQA Prompt and Judgment |
| --- |

**Prompt/Question**

Who was the last man to walk on the Moon?

**Response A: TinyLlama/TinyLlama-1.1B-Chat-v1.0**

Neil Armstrong

**Response B: HuggingFaceTB/SmolLM3-3B**

<think> Okay, so I need to figure out who the last man to walk on the Moon was. Let me start by recalling what I know about the Apollo missions. The Apollo program was NASA's effort to land humans on the Moon, right? The first moon landing was Apollo 11 in 1969, with Neil Armstrong and Buzz Aldrin. Then there were several more Apollo missions, but after that, the program ended. Wait, but I think there were more Apollo missions after Apollo 11. Let me think. Apollo 12, 13, 14, 15, 16, 17, and 18. Apollo 13 was the one where they had to abort the mission because of an oxygen tank explosion. Apollo 14 had Alan Shepard as commander. Then Apollo 15, 16, 17, and 18. Now, the last mission would be Apollo 17, right? Because after that, the program was supposed to end. Let me check the dates. Apollo 17 was in 1972.
So if Apollo 17 was the last one, then the last man to walk on the Moon would be from that mission. Who were the astronauts on Apollo 17? I think it was Eugene Cern.

**Judgment**

Response B [HuggingFaceTB/SmolLM3-3B] is better.

*Table 10.* Query template for the reading comprehension task (SQuAD 2.0).

```
You are an expert evaluator for reading comprehension and factual question
answering.  Compare two responses (A and B) to the same question.  Which response
is better overall based on:
1.  Factual Correctness:  Is the answer factually correct given general
knowledge?
2.  Relevance:  Does it directly address the question asked?
3.  Clarity & Coherence:  Is the response clear and logically presented?
You MUST output only a single character:  'A' or 'B'.

Query:  <user_query>

Response A: <response_a>
Response B: <response_b>

Comparison:  <one-sentence comparison >
Preferred:  "A" or "B"
```

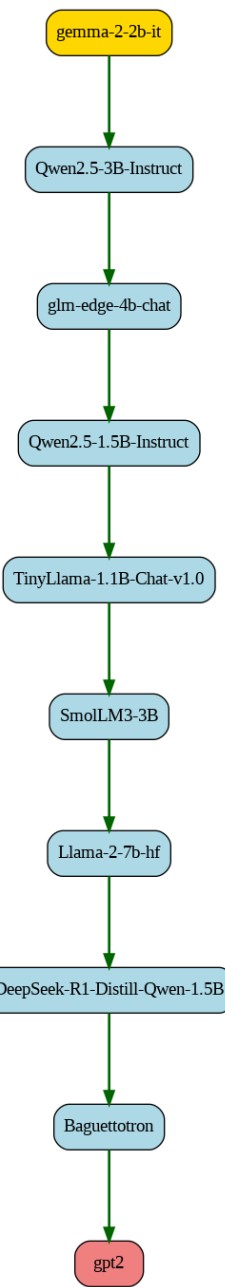

*Figure 14.* DAG of LLMs ranking over databricks-dolly-15k dataset when $t = 5000$. Arrow $A \rightarrow B$ indicates model A outperforms model B.

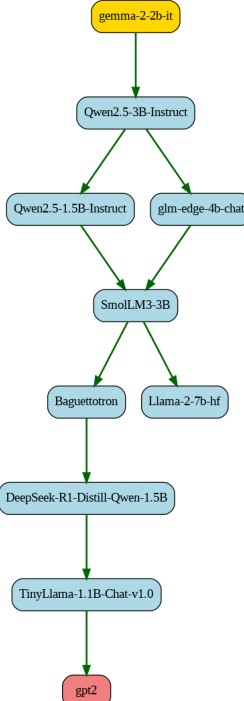

*Figure 15.* DAG of LLMs ranking on the MMLU multiple-choice dataset when $t = 5000$. Arrow $A \rightarrow B$ indicates model A outperforms model B.

*Table 11.* Sample Responses of the reading comprehension task (SQuAD 2.0) Prompt and Pairwise Judgment

| Sample Responses of SQuAD 2.0 Prompt and Judgment |
| --- |
| **Prompt/Question** |
| What does MP3 use to format its data? |
| **Response A: PleIAs/Baguettotron** |
| MP3 format uses a specific type of data structure called a "data block". 
 Here is the breakdown: 
 Data Block: A data block is a block of data that is organized in a specific way. 
 Example: The data block for "10" is a block of 10 characters. 
 Example: The data block for "10" is a block of 10 characters. 
 In short, the "data block" is the basic unit of data storage for MP3. It's the smallest piece of data that can be stored together to form a complete file. |
| **Response B: zai-org/glm-edge-4b-chat** |
| MP3 files are formatted using a combination of lossy compression algorithms to reduce the file size while retaining an acceptable level of audio quality. The specific algorithm used in MP3 compression is called the MPEG Audio Layer III compression, which is part of the MPEG-1 and MPEG-2 audio coding standards developed by the Moving Picture Experts Group (MPEG). 
 The MPEG Audio Layer III compression works by encoding the audio signal in such a way that the most perceptually significant parts of the audio are retained, while less important information is discarded. This process is achieved through a combination of frequency masking, psychoacoustic modeling, and the use of a 32-bit floating-point filter bank. 
 The MP3 file format itself is defined in the MPEG-4 Part 14 standard, which is commonly referred to as the MP4 container. The MP4 container supports various types of multimedia data, including audio, video, and text. MP3 files are typically stored within this container, but they can also be found in other formats, such as .wav, .aac, or .ogg, depending on the specific requirements and applications. |
| **Judgment** |
| Response B [zai-org/glm-edge-4b-chat] is better. |

*Table 12.* Query template for the instruction-following and text generation task.

```
You are an expert evaluator for instruction-following and text generation.
Compare two responses (A and B) to the same user instruction.  Which response
is better overall based on:
1.  Instruction Following:  Does it correctly do what the instruction asked?
2.  Coherence & Helpfulness:  Is the response logical, clear, and useful?
3.  Language Quality:  Is it well-written with good grammar and style?
You MUST output only a single character:  'A' or 'B'.

Query:  <user_query>

Response A: <response_a>
Response B: <response_b>

Comparison:  <one-sentence comparison >
Preferred:  "A" or "B"
```

*Table 13.* Sample Responses of an Instruction-Following and Text Generation Prompt and Judgment

**Sample Responses of an Instruction-Following and Text Generation Prompt and Judgement**

**Prompt/Instruction**

Given a reference text about The Universal Data Element Framework (UDEF), tell me how the framework is used.

**Response A: TinyLlama-1.1B-Chat-v1.0**

The Universal Data Element Framework (UDEF) is a software-defined data model that enables interoperability between different data sources. It allows organizations to create standardized data structures that can be used across multiple applications and systems. The framework is used to ensure that data is consistent, accurate, and up-to-date, which is crucial for efficient data-driven decision making.

**Response B: gemma-2-2b-it**

The Universal Data Element Framework (UDEF) is a powerful tool for building interoperability across various healthcare systems and organizations. It's essentially a standardized vocabulary of data elements that can be used to describe medical information. Here's how it's used:
1. **Standardization and Interoperability**:
**Consistent Data Representation**: UDEF provides a common language for describing medical data, ensuring that different systems and organizations can understand and exchange information regardless of their specific software or technology.
**Data Exchange**: UDEF facilitates the exchange of medical data between different healthcare systems, such as Electronic Health Records (EHRs), lab information systems, and pharmacies.
**Data Integration**: It allows for the integration of data from multiple sources, creating a unified view of patient information for better clinical decision-making.
2. **Improved Data Quality and Accuracy**:
**Controlled Vocabulary**: UDEF's standardized vocabulary helps to ensure consistent and accurate data encoding, reducing errors and improving data quality.
**Data Validation**: The framework includes rules and guidelines for data validation, ensuring that data is entered correctly and meets specific standards.

**Judgment**

Response B [gemma-2-2b-it] is better.

*Table 14.* Query template for the multiple-choice reasoning task (MMLU).

```
You are a strict multiple-choice question evaluator.  You will see a
multiple-choice question and responses from two models.  Judge which model's
response is better (more accurate, clearer, more logical).  You MUST output
ONLY 'A' or 'B', no other content is allowed.  - Output 'A' if the first model's
response is better - Output 'B' if the second model's response is better

Query:  <user_query>
A. <user_query>
B. <user_query>
C. <user_query>
D. <user_query>
Please select the correct answer (A/B/C/D):

Response A: <response_a>
Response B: <response_b>

Comparison:  <one-sentence comparison >
Preferred:  "A" or "B"
```

*Table 15.* Sample Responses of the multiple-choice reasoning task (MMLU) Prompt and Pairwise Judgment

| Sample Responses of an MMLU Prompt and Judgment |
| --- |

**Prompt/Question**

What dietary pattern shows most promise in preventing age-related cognitive decline?
A. The Western diet
B. The Mediterranean diet
C. Neither
D. Both

**Response A: google/gemma-2-2b-it**

The correct answer is **B. The Mediterranean diet**.
This dietary pattern emphasizes fruits, vegetables, whole grains, healthy fats, and lean proteins, which have been widely associated with better cognitive outcomes and reduced risk of age-related decline.

**Response B: TinyLlama/TinyLlama-1.1B-Chat-v1.0**

D

**Judgment**

Response A [google/gemma-2-2b-it] is better.

