# OpenReview forum: "Anytime-Valid Inference for Online Ranking of Large Language Models"
_ICML.cc/2026/Conference — ICML 2026 regular_

### Official Review · Reviewer_2Uv6 · 2026-03-08

**Soundness:** 3
**Presentation:** 4
**Significance:** 2
**Originality:** 3
**Overall Recommendation:** 5
**Confidence:** 4

**Summary:**

This paper builds on ongoing work in anytime-valid testing and e-processes to construct anytime-valid ranking of LLMs. The authors consider the setting where there is a large set of models, and a stream of pairwise comparisons on various inputs (prompts) x, where users are able to rate if model A or model B is preferred on that input. They then set up a null, where for each pair (j,k), they reject the null k > j (in ranking) once they've accumulated enough evidence via an e-process in favor of j > k.

There are two versions of the e-process used. The first is "covariate-agnostic", where the binary preferences are revealed as a stream and there is no explicit conditioning on x. The second is covariate-aware, where the authors train a secondary Bradley-Terry model and use this to set the priors over $p_{j > k}$. There is also an introduction of a tournament-based sampling strategy, where model pairs with greater uncertainty are prioritized.

The authors experiment with GPT as the preference annotator on several open-weight language models.

**Compliance With Llm Reviewing Policy:**

Affirmed.

**Final Justification:**

My primary concern was around transitivity and the authors explained that transitivity is only necessary for pruning.

**Key Questions For Authors:**

Q1: What is $\theta$ initialized to in Eq. 3, before any samples have been allocated to (j,k)?

Q2: Could you please report some transitivity metrics. In particular, please report if there are any cycles in your preference datasets. Please also discuss how crowdsourced annotations could potentially break transitivity, unlike GPT-4o-mini.

Q3: Would any of your results change if you introduced the starvation-free policy discussed in weaknesses?

**Limitations:**

Yes.

**Strengths And Weaknesses:**

Strengths:
S1: The paper is well-written and easy to follow. The authors clearly state all assumptions, define notation, and survey relevant literature.

S2: The empirical benefits of the method are clear, especially in Table 2. For instance, SERPANT is able to recover at least 80% of the ordering on all four datasets tested using 5000 samples.

S3: The contribution is clear. E-processes are a natural fit to crowdsourced leaderboards. Indeed, I believe the original chatbot arena paper (https://arxiv.org/abs/2403.04132) mentioned using e-processes for sample allocation ("Armed with this data, we employ [...] E-values of Vovk & Wang (2021), to estimate the ranking over models as reliably and sample-efficiently as possible") but did not specify how they did it.

Weaknesses:
W1: Assumption of transitivity. Although I understand that transitivity is a necessary assumption for the theory to hold, this assumption may not hold in practice. It is possible, especially in crowdsourced annotations, that with three models A, B, and C, A is preferred to B (on average), B to C, and C to A. I would prefer the authors include some discussion on this. For instance, they could verify in their databricks-dolly-15k dataset that there are no such cycles.

W1.5: Related to the above, the authors use GPT-4o-mini as the preference annotator. This makes the transitivity assumption more reasonable here, but it may not generally hold in crowdsourced settings (where there isn't a single annotator with consistent ratings).

W2: It is unclear to me that stopping the allocation of samples to models outside the top k is desirable, when there is some chance of an error. As a model developer, "starvation-free" allocations seem desirable (i.e., with some $\varepsilon$ probability, I will always be sampled, even if I am outside the top $k$). Why should I submit my model to a leaderboard with $\alpha=0.1$, where there is a 10% chance of eventually being starved of samples even when I am in the top k?

W3: Although the re-parameterization of the prior is attractive in Eq. 4 appears attractive, it seems to be (as written) invariant to the value of $n_t$, instead solely being a function of $\hat{p}_t$. I believe this may lead to some power issues, as you could concentrate the beta distribution (i.e., reduce variance) for larger $n_t$.

W4: The method is empirically very conservative, achieving zero FWER at a nominal $\alpha=0.1$ (Fig. 3).

---

> ### Author Rebuttal · Authors · 2026-03-29
>
> We thank Reviewer 2Uv6 for the thoughtful comments and for highlighting transitivity and related questions.
>
> Weakness 1: Transitivity.
>
> Thank you for raising this important concern. We agree that strong non-transitivity may arise for some specific cases. We would like to clarify the following.
>
> First, our procedure does not fundamentally require a transitivity assumption for validity. Transitivity is used only to support the pruning step and improve efficiency. If one does not wish to assume transitivity, the pruning step can simply be removed from the algorithm. The resulting procedure remains a valid multiple-testing procedure with rigorous FWER control. It evaluates each pair independently, outputting a directed graph of significant pairwise preferences. Any cycles are faithfully captured with high probability rather than forced into a hierarchy. In other words, the statistical guarantee is provided by the sequential testing framework itself, not by the pruning heuristic.
>
> Second, in many ranking settings, especially for tasks with scalar evaluation scores such as summarization, coding, or mathematics, transitivity is a natural and widely adopted structural assumption. More broadly, when users request a ranking from best to worst, this already implicitly assumes a transitive ordering. Our pruning step is designed for precisely this regime, where a global ranking is a meaningful target and transitivity is a reasonable approximation.
>
> In the revision, we will make clear that transitivity is an optional structural assumption used only for pruning, not for validity.
>
> Key Question 1: Initialization of $\theta$.
>
> Thanks for the question. In Eq. 3, before any samples have been allocated to pair $(j,k)$, we initialize $\theta_{jk}=0$. Under the Bradley–Terry model, this corresponds to equal win probabilities for the two items. In other words, the pair is treated as equally likely a priori until comparison data are observed.
>
> In the revision, we clarified that $\theta_{jk}=0$ is initialized to 0.
>
> Key Question 2 and Weakness 1.5: Transitivity validation.
>
> We use the Databricks setting to assess the benefit of pruning via transitivity and to examine whether the required condition holds under the same setup as in the main paper. As expected, disabling pruning results in some power loss, but we do not find any contradiction.
>
> | Dataset | Transitivity | t=1000 | t=2000 | t=3000 |
> |---|---|---:|---:|---:|
> | **Databricks** | Prune by transitivity | **28 (62%)** | **41 (91%)** | **43 (96%)** |
> |  | Without Prune | 24 (53%) | 35 (78%) | 38 (84%) |
>
>
> Key Question 3 and Weakness 2: Starvation-free policy
>
> Thank you for this thoughtful comment. We agree that starvation-free allocation may be preferable in some practical deployments, since it guarantees that models outside the current estimated top-k set still retain a nonzero probability of being sampled.
>
> One strength of the e-process framework is its substantial flexibility in the sampling policy. Our proposed top-k procedure is therefore not limited to a hard truncation rule. It can be extended by adding a resampling component: at each stage, a model outside the current top-k set is given a small probability of being compared with the currently worst model in the top-k set. In this way, every model remains eligible for future sampling, while the procedure continues to focus primarily on the most relevant ranking boundaries.
>
> This starvation-free modification does not affect the main validity guarantees of our framework; rather, it represents an implementation variant within the same general methodology. In the revision, we have added this discussion and clarified that the proposed framework can naturally accommodate such starvation-free policies.

---

> > ### Author Rebuttal · Reviewer_2Uv6 · 2026-04-02
> >
> > Thank you for the additional comments and for the detailed thoughts on transitivity. I will update my score to an "accept" and encourage the authors to discuss transitivity in greater depth.

---

### Official Review · Reviewer_TrRW · 2026-03-09

**Soundness:** 2
**Presentation:** 3
**Significance:** 3
**Originality:** 3
**Overall Recommendation:** 4
**Confidence:** 3

**Summary:**

The paper focuses on the anytime-valid inference problem in online ranking for large language models (LLMs). Addressing the limitations of classical statistical inference caused by adaptive sampling and continuous monitoring in existing LLM online evaluation, it proposes a novel framework called SERPANT, which provides the first theoretically rigorous anytime-valid error control for LLM online ranking. Additionally, it designs an efficient adaptive sampling strategy and achieves the construction of anytime-valid confidence sets for top-k optimal models.

**Compliance With Llm Reviewing Policy:**

Affirmed.

**Final Justification:**

Based on the author's two responses, I have understood their understanding of transitivity and resolved my confusion about method complexity. I am now willing to accept this article

**Key Questions For Authors:**

1. Regarding assumption, the prompt mentioned in line150 is i.i.d. How to explain this and whether it is reasonable.
2. Furthermore, is it reasonable for the article to assume that preferences have transitivity? In real scenarios, there may be situations where A>B, B>C but C>A. Is the transitivity pruning of Algorithm 1 affected by the order of data input? How can fairness and effectiveness be guaranteed if the derived preference relationship has not been directly compared and verified? Why not directly test all models to ensure stable and fair results?
3. Can this framework maintain an online version, such as how to maintain a new batch of data or a new batch of models, or does it require a complete re inference
4. At present, if the overall complexity of this framework is for scenarios with more than 100 models, K>10^5， It requires an extremely large annotated sample to make the e-value exceed the threshold, and it may not be possible to confirm a large number of true preference relationships under limited budget. Can this method still be used in this case?
5. DAG is a good representation of preference relationships, and I would like to know if this result can be represented and integrated in multiple scenarios

**Limitations:**

yes

**Strengths And Weaknesses:**

Strengths:
1. The article starts from the perspective of statistical inference and has a strong theoretical foundation.
2. The framework design is very complete, and the evaluation result of DAG is a very valuable result

Weaknesses:
1. The universality of the core assumptions is insufficient.
2. The efficiency and feasibility in large-scale scenarios have not been validated
3. Relying on the reliability of automatic evaluators, the deviation risk in real scenarios has not been fully addressed

---

> ### Author Rebuttal · Authors · 2026-03-29
>
> We thank Reviewer TrRW for the thoughtful comments and for highlighting transitivity and related assumptions.
>
> Key Question 1: i.i.d. Assumptions.
>
> We thank the reviewer for raising this important point. We agree that the i.i.d. assumption may not hold exactly in all real-world systems. However, it is a reasonable approximation for online evaluation settings and is widely adopted in the online learning literature. Our experiments were designed to closely match this setting, and we will clarify this in the revision.
>
> First, in the online experimental design, $P_x$ represents the distribution of incoming prompts. When prompts are randomly sampled from this distribution and evaluated separately, the assumption is a reasonable approximation. In particular, $P_x$ can be viewed as the distribution induced by the full pool of potential users in an online evaluation system. Each prompt is then treated as a draw from this common distribution, making the i.i.d. assumption a reasonable approximation.
>
> Second, in our numerical experiments, we designed the evaluation pipeline to satisfy this assumption. For each pairwise comparison, we independently sampled a prompt from the dataset.
>
> Key Question 2: Transitivity.
>
> Thank you for raising this important concern. We agree that strong non-transitivity may arise for some specific cases. Our procedure does not fundamentally require a transitivity assumption for validity. Transitivity is used only to support the pruning step and improve efficiency. In addition, we use the Databricks setting to assess the benefit of pruning via transitivity and to examine whether the required condition holds under the same setup as in the main paper. As expected, disabling pruning results in some power loss, but we do not find any contradiction. Please see our response to Key Question 2(1) and Key Question 2(2) from Reviewer wgk6 for details.
>
> Key Question 3: Online.
>
> Thank you for this important question. The framework is designed precisely for online evaluation, and therefore does not require complete re-inference whenever new data arrives.
>
> For a new batch of comparison data, the procedure is updated sequentially: the e-process is defined recursively and continues to accumulate evidence as additional observations are collected. Thus, new data can be incorporated directly into the existing inference pipeline, while preserving validity at every stopping time. In this sense, the method is naturally suited to continuous online evaluation.
>
> For a new batch of models, the situation is slightly different. Existing pairwise inferences among previously included models do not need to be recomputed from scratch. Instead, one only needs to initialize and maintain the new pairwise comparisons involving the newly added models, allocating additional budget to the error rate. The previously accumulated evidence for old model pairs can be retained because the e-value is valid at every stopping time. This is still an incremental update, not a full restart.
>
> Therefore, the framework supports online maintenance in both senses. In the revision, we will clarify more explicitly how the procedure supports online updates for both newly arriving data and newly introduced models, without requiring complete re-inference from scratch.
>
> Key Question 4: Larger models.
>
> Response: Thank you for this important question. We agree that the rejection threshold becomes more stringent as the number of models grows, so practical scalability is an important consideration. That said, the e-process is an exponential-evidence procedure: when two models exhibit genuinely different behavior, the corresponding e-value typically grows exponentially fast with the number of comparisons. As a result, for pairs with a non-negligible performance gap, the ordering can still be identified within a reasonable comparison budget, even under a multiplicity-adjusted threshold.
>
> In our current experiments, due to computational constraints, we study up to 50 models, totaling 20,000 comparisons. For context, Text Arena reports 330 models and 5,602,397 votes as of March 20, 2026, which suggests that comparison budgets at this scale are feasible in real online arenas. We therefore believe the proposed method is practically viable, while evaluation at even larger scales remains an important direction for future work.
>
> Key Question 5: DAG representation.
>
> Thank you for this helpful comment. We agree that a DAG is a natural representation of preference relationships, since it captures partial orderings without forcing a total ranking. In our framework, the inferred pairwise comparisons can indeed be summarized as a DAG. This perspective is also useful in multi-scenario settings: one may build separate DAGs for different task classes or user groups, or define a mixture over scenarios to obtain an aggregate DAG. In the revision, we will make this interpretation more explicit and discuss how the framework can be integrated across multiple scenarios.

---

> > ### Author Rebuttal · Reviewer_TrRW · 2026-04-03
> >
> > I'm sorry I didn't see any explanation about transitivity in your reply. I am still curious whether pruning under different initial conditions under current algorithms will yield the same partial order results, which is important for fairness. Furthermore, I do not understand why 50 models (20k dataset) and 300+models (5000k dataset) are at the same level, and I still question the complexity and universality of large-scale data (real-world scenarios)

---

> > > ### Author Response · Authors · 2026-04-04
> > >
> > > Thank you for the follow-up. We apologize that our previous reply did not clearly explain the role of transitivity. As requested by multiple reviewers, we had referred to related responses elsewhere, but we agree that this point should be stated directly and more clearly here. Below, we restate our explanation of transitivity in more precise terms.
> > >
> > > * Validity does not depend on transitivity. The core of our method is the sequential pairwise testing procedure, which alone provides the anytime-valid family-wise error rate (FWER) guarantee. Importantly, this component is entirely independent of any transitivity assumption: its statistical validity holds regardless of the underlying preference structure.
> > >
> > > * Transitivity as an efficiency-enhancing assumption (pruning). The pruning step is introduced solely to improve ranking efficiency. Because its purpose is to accelerate the construction of a linear ranking (e.g.,1,2,3), and such a ranking implicitly relies on transitivity. For many objective tasks, such as coding or mathematical reasoning, this is a reasonable assumption.
> > >
> > > We acknowledge that transitivity may not always hold, particularly in more subjective tasks. If a practitioner does not wish to assume transitivity, the pruning step can simply be dropped without affecting the FWER guarantee. In the absence of the transitivity assumption, the task naturally shifts from producing a strict linear ranking to producing a directed graph representing the true preference relations. If the underlying oracle exhibits cyclic preferences (e.g., A > B > C > A), our unpruned method will faithfully preserve and output exactly these cyclic relations with high probability.
> > >
> > > The concern about different initial conditions leading to different final results (i.e., path dependency or "fairness") is completely resolved by how the algorithm handles the transitivity assumption. We can break this down into two distinct scenarios:
> > > *  When the transitivity assumption holds. Because the core sequential tests rigorously control the FWER, the pairwise preference relations we establish are highly reliable. As a result, apart from the small error probability already controlled by the FWER, the algorithm will recover the same final partial order regardless of the initial conditions or the order in which pairs are evaluated. In this setting, pruning does not create a fairness concern.
> > > * The transitivity assumption does NOT hold. You are absolutely correct that a fairness issue would arise if there were true cycles in the data, but one would still choose to apply the pruning step. In this case, the outcome would indeed depend on the initial conditions: whichever pairs are tested first would determine which items are unfairly pruned later. However, this scenario represents a fundamental misspecification of the model. In this case, we recommend dropping the pruning step if the user truly believes that transitivity does not hold. Without pruning, the procedure relies solely on the sequential pairwise tests; since the FWER is controlled, the resulting set of discovered pairwise relations remains statistically valid regardless of the initial conditions or the order in which pairs are evaluated.
> > >
> > > We would like to clarify our previous statement regarding the scale of these datasets. Our point was that “50 models / 20k comparisons” and “300+ models / 5M comparisons” represent a similar level of difficulty for our algorithm.
> > >
> > > * Enough pairwise comparisons in practice. Because our framework is built on sequential pairwise tests rather than a single global ranking statistic, its effective complexity is determined by the number of comparisons per pairwise hypothesis. In our simulation study, we use 50 models and 20k comparisons to illustrate the effectiveness of the proposed method. We would like to clarify that, on practical platforms such as LMArena, the volume of pairwise comparisons is typically sufficient to support the effective use of our method in real-world settings.
> > >
> > > * Effectiveness of e-values in large-scale settings. Once a pair exhibits a clear performance gap, the corresponding evidence, as measured by the e-value, grows exponentially. For example, if Model A wins 75 of 100 comparisons against Model B, the e-value is already $1.09×10^5$. To maintain valid FWER control, the rejection threshold grows only quadratically with the number of models. Consequently, for clearly separated pairs, only a very small increase in the number of comparisons is needed to overcome the higher threshold even in larger-scale settings.
> > >
> > > * Pruning and tournament. We acknowledge that as the number of models increases, the total number of possible pairs also grows quadratically. However, with additional structural assumptions, such as transitivity, we can avoid exhaustively testing all pairs. Thus, the combination of pruning and tournament-based allocation makes the method practically scalable and broadly applicable to large-scale real-world ranking scenarios.

---

### Official Review · Reviewer_tkDt · 2026-03-09

**Soundness:** 2
**Presentation:** 3
**Significance:** 3
**Originality:** 3
**Overall Recommendation:** 3
**Confidence:** 4

**Summary:**

The authors devise a new approach to efficiently sampling preferences in an effort to construct a valid ranking of models utility. Specifically, the paper invokes careful anytime valid / martingale analyses to analyze their novel means of adaptive sampling (within the active learning paradigm) for estimation of preferential ranking efficiently. This framework, denoted Sequential E-value Ranking and Pruning via Adaptive Null Testing (SERPANT) considers model comparison to be a pairwise hypothesis test with an error rate controllable at any stopping time. Beyond the analytic justifications, the authors further invoke a novel tournament-based sampling strategy to adaptively select informative comparisons to evaluate, empirically showing that this improves significantly on random samplings.

**Compliance With Llm Reviewing Policy:**

Affirmed.

**Key Questions For Authors:**

It seems limited to marginalize the null-hypothesis formulation over $\mathcal{P}_X$?

Why not consider larger models in the analysis?

Was the LLM judge certified to exhibit transitivity in the experimental results?

**Limitations:**

The major limitation is using LLM as a judge. I encourage the authors to comment on this point and address my concerns. I am happy to increase my score if this is properly addressed as, once again, I find the theoretical contributions to be compelling.

**Strengths And Weaknesses:**

# Strengths
The main strength of this work is the theoretical contributions. The tournament sampling and anytime-valid inference are elegantly presented to emphasize efficient means of reconstructing a preferential ranking. This is especially relevant for the case of human judgements which can be prohibitively costly--any means of reducing the number of such labelings needed is highly impactful for studies that leverage these pair-wise annotations.

Additionally, the paper is written in a clear manner. The appendix proofs appear largely correct apart from a few proof components which may need more elaboration (see Questions and Weaknesses).

# Weaknesses
While the theoretical results are compelling, I think the motivation and empirical results are confusing and a serious deterrent of the current submission.

- The most glaring contradiction seems to be the empirical sections reliance on an LLM-judge. The authors motivate their study and the entire sequential framework by citing the need to reduce costly human annotations, but the experiments use GPT to judge (which circumvents this cost already). I think this is a serious hindrance. The paper could instead be reframed as more efficiently sampling an LLM's preference but I think that introduces a cascade of issues since such models are inherently very noisy and crucially **are not guaranteed to exhibit transitivity**.
Fundamentally, I think it is hard to certify the merits of this method for the given model preference ranking without a ground-truth to compare against.

- Extending on this, I think the efficiency gains would be more impactful if compared against an empirical baseline preference dataset. Based on the above, I think it would further be insightful to empirically validate the transitivity of LLM preference when used as a judge.

- The authors generate DAGs for inferred partial orders on several datasets. But these results seemingly have no bearing unless cross-referenced against public leaderboards (like Arena ELOs)? If SERPANT is significantly different than the heuristic consensus, the authors should discuss why this might be the case. If the rankings agree with these public leaderboards, that is also very insightful.

- Moreover, I am confused by the utility of deriving a ranking of LLMs in the first place. Correct me if I am wrong, but such rankings are fuzzy and vary between persons and tasks. Thus, efficiently deriving such a ranking would need to be done for each person and then for each task for that person. It seems, for this motivating problem, heuristics may be better.

- There are potential flaws in the proof of Theorem A.3 (please clarify these if I am wrong). The formulation of the prior $\hat{g_t}$ relies on the current covariant which seemingly violates the strict $\mathcal{F_{t-1}}$ measurability. This measurability is necessary to factor the expectation and maintain the martingale property. Later in this proof, it appears that the integral of a product is equated with a product of integrals in recovering $M_{t-1}(p_0)$. These flaws appear to break the FWER guarantees for the covariated-assisted extension.

Overall, the theoretical contributions are interesting but I believe they would be best applied to a more natural fit problem.

---

> ### Author Rebuttal · Authors · 2026-03-29
>
> We thank Reviewer tkDt for the thoughtful comments and for highlighting transitivity and related questions.
>
> Weaknesses 1(1): Annotation cost.
>
> Thanks for raising this concern. Pairwise comparison-based evaluation is often much less expensive than building a fixed benchmark dataset, because annotators need only make relative judgments. In contrast, fixed benchmarks require substantial effort to define evaluation criteria. Finally, many important LLM qualities, such as helpfulness, are difficult to encode in a fixed dataset but can be naturally captured through pairwise comparisons.
>
> The use of LLMs as judges is primarily intended as a practical proxy for human preferences. We will clarify this point in the revision.
>
> Weaknesses 1(2): Transitivity.
>
> Thank you for raising this important concern. We agree that strong non-transitivity may arise for some specific cases. We would like to clarify the following. Our procedure does not fundamentally require a transitivity assumption for validity. Transitivity is used only to support the pruning step and improve efficiency. Please see our response to Key Question 2(1) from Reviewer wgk6 for details.
>
> Weaknesses 2 and Key Question 3: Against an empirical baseline preference dataset.
>
> We use the Databricks setting to assess the benefit of pruning via transitivity and to examine whether the required condition holds under the same setup as in the main paper. As expected, disabling pruning results in some power loss, but we do not find any contradiction.
>
> | Dataset | Transitivity | t=1000 | t=2000 | t=3000 |
> |---|---|---:|---:|---:|
> | **Databricks** | Prune by transitivity | **28 (62%)** | **41 (91%)** | **43 (96%)** |
> |  | Without Prune | 24 (53%) | 35 (78%) | 38 (84%) |
>
> Weaknesses 3: Public leaderboards.
>
> Thank you for raising this point. We compare 10 models selected from the Arena leaderboard for text tasks against the ranking based on empirical win rates, which lacks error-rate control. Our ranking is broadly consistent with the leaderboard, with only one disagreement. This discrepancy may be explained by the fact that our evaluation is conducted on Databricks questions. The empirical approach incurs a non-negligible number of false discoveries throughout.
>
> | Method | Metric | t=1000 | t=2000 | t=3000 | t=4000 |
> |---|---|---:|---:|---:|---:|
> | **SERPANT** | true discovery | 16 (36%) | 29 (64%) | 32 (71%) | 32 (71%) |
> |  | false discovery | 0 (0%) | 0 (0%) | 1 (2%) | 1 (2%) |
> | **Empirical** | true discovery | 39 (87%) | 40 (89%) | 40 (89%) | 41 (91%) |
> |  | false discovery | 6 (13%) | 5 (11%) | 5 (11%) | 4 (9%) |
>
> Weaknesses 5: $\hat{g}_t$.
>
> Thank you for carefully reviewing Theorem A.3 and for pointing out this issue. We agree that $\hat{g}_t$ should not depend on the current covariate. This is not our intention. Our definition of $\hat{g}_t$ depends only on the averaged probability computed from the historical data available up to time $t$ on Line 208. Therefore, the weights depend on the average score based on past data, and the required measurability condition holds.
>
> We apologize for the potentially misleading notation. In the revision, we will replace $T_{t}$ by $T_{t-1}$ to make clear that the construction depends only on past information.
>
> Weaknesses 4 and Key Question 1: Marginalization.
>
> Thank you for raising this important point. We agree that LLM rankings are inherently task- and population-dependent. In some sense, marginalizing over $P_X$ is not a limitation, but the formal definition of the population with respect to which ranking is performed. Conceptually, each user facing a specific task and prompt may have their own subjective ranking of model outputs. However, such a prompt- and user-specific ranking is useful only in that narrow setting, offers little guidance to others, and is generally not estimable in practice because data at that level are unavailable. In contrast, a practically useful ranking should provide guidance for a broader population of users and tasks. For instance, if the goal is to rank models for assisting academic writing, the relevant users may come from different disciplines and use different prompts. Since neither the future user nor the prompt is known in advance, a natural target is a marginalized ranking that averages over the user and prompt distribution of interest. Such a target is both estimable and practically useful, and it can still provide meaningful guidance for subgroups with similar preferences or related tasks.
>
> In the revision, we will clarify this point more explicitly.
>
> Key Question 2: Larger models
>
> Response: Thank you for this suggestion. Our current analysis focuses on a moderate set of models of moderate size to illustrate the main methodological contribution and to keep the experimental study computationally manageable. Our method is not restricted to small or medium-sized models. In the revision, we will explicitly state that the current model scale was chosen for experimental tractability.

---

> > ### Author Rebuttal · Reviewer_tkDt · 2026-04-03
> >
> > Hi, thank you for the response and expansion on my critiques. I still find the utilization of LLM as a judge, in combination with the transitivity issue (also noted by other reviewers), as issues with the work and will keep my score as is.

---

> > > ### Author Response · Authors · 2026-04-04
> > >
> > > Thank you for your continued engagement with our work.  We would like to clarify the fundamental mechanics of our method, which remain completely valid regardless of the evaluator’s identity or the presence of transitivity.
> > >
> > > 1. Our paper motivates the need for efficient sequential testing to reduce the cost of human annotations, which is widely recognized as a major bottleneck in RLHF and model evaluation. Because conducting large-scale, sequential human annotation is logistically and financially infeasible for a small academic team, we follow the standard, widely accepted practice in the literature of using an LLM-as-a-judge as a proxy to validate the statistical properties of the algorithm. Crucially, our theoretical framework is entirely agnostic to the source of the preferences. The statistical validity of our procedure is independent of the identity of the evaluator; the theoretical guarantees hold whether the preferences are provided by a human or an LLM. Our experiments serve to prove that the algorithm efficiently and accurately uncovers the underlying preference distribution of the chosen oracle, demonstrating its practical viability for future human-in-the-loop deployments.
> > >
> > > 2. You raise a concern about transitivity violations in LLM preferences. We wish to highlight three critical points regarding transitivity:
> > >
> > > (a) FWER Guarantees are Independent of Transitivity: This is the most important technical point. If we do not wish to rely on the transitivity assumption, we can simply remove the pruning step from our algorithm. The Family-Wise Error Rate (FWER) guarantee holds entirely independently of the pruning step.
> > >
> > > (b) Faithful Recovery of Preference Graphs: If the underlying oracle (human or LLM) exhibits cyclic preferences (e.g., A > B > C > A), our unpruned method will faithfully preserve and output exactly these cyclic relations with high probability. In the absence of transitivity, it naturally shifts from outputting a strict linear ranking to outputting a directed graph representing true preference relations. Our algorithm correctly recovers this directed graph with rigorous statistical guarantees.
> > >
> > > (c) Transitivity is Domain-Dependent, not Evaluator-Dependent: We acknowledge that strict transitivity does not always hold. However, neither LLMs nor humans can guarantee transitivity in subjective tasks. Conversely, if we restrict our attention to objective reasoning tasks (such as mathematics or coding), it is highly reasonable to assume that transitivity holds for both humans and LLMs.
> > >
> > > In summary, removing the pruning step yields a method that strictly controls FWER for directed preference graphs without requiring transitivity, while keeping it yields an efficient ranking algorithm for domains where transitivity is a safe assumption. Both the theoretical contributions and the empirical validation of the FWER control remain sound. We hope this clarifies the robustness of our framework.

---

### Official Review · Reviewer_wgk6 · 2026-03-10

**Soundness:** 3
**Presentation:** 3
**Significance:** 3
**Originality:** 3
**Overall Recommendation:** 5
**Confidence:** 3

**Summary:**

This paper tackles a major flaw in how we currently rank LLMs online. Right now, platforms monitor results and stop testing when a winner looks clear, which completely ruins standard statistical math and inflates false discoveries. The authors fix this with SERPANT, a framework that uses e-processes to guarantee the error rate (FWER) stays strictly controlled no matter when you stop testing. They also introduce a "tournament sampling" method that actively picks the hardest matchups to save money on expensive human labeling.

**Compliance With Llm Reviewing Policy:**

Affirmed.

**Final Justification:**

I am satisfied with the author's rebuttal and recommend the article "Accept".

**Key Questions For Authors:**

How does SERPANT handle identical models? Since the framework structurally assumes $C_{j,k} \in \{1, -1\}$, won't the e-process simply run forever and drain the evaluation budget if two models tie? Is there a planned stopping rule for equivalence?

How vulnerable is the pruning step to multi-dimensional LLM skills?

Does the $K = m(m-1)/\alpha$ threshold scale practically to real-world arenas?

**Limitations:**

See Weakness

**Strengths And Weaknesses:**

Strengths

It directly solves the "data peeking" problem that plagues live, continuous leaderboards.

The e-process foundation is elegant.

The tournament sampling strategy is a highly practical engineering touch.

Weaknesses

The mathematical model explicitly assumes there are no ties ($C_{j,k} \in \{1, -1\}$). Real LLMs often perform at the exact same level. If two models are truly tied, the e-process will never grow enough to cross the rejection threshold.

The system automatically prunes tests based on transitivity (if A beats B, and B beats C, then it assumes A beats C). LLM capabilities are highly multi-dimensional. Model A might beat B at coding, B beats C at math, and C beats A at writing. If this "rock-paper-scissors" loop happens in the data, the hard-coded pruning will create false rankings and immediately break the rigorous FWER guarantees the paper relies on.

To control the global error rate, the rejection threshold is set to $K = m(m-1)/\alpha$. This is a Bonferroni-style correction that scales quadratically. The paper tests up to 50 models, but real arenas have well over 100.

---

> ### Author Rebuttal · Authors · 2026-03-29
>
> We thank Reviewer wgk6 for the thoughtful comments and for highlighting ties and transitivity in the ranking problem.
>
> Key question 1(1): Identical models.
>
> Thank you for raising this very important point. We totally agree that ties are important in realistic LLM evaluation, and we would like to clarify that our procedure can accommodate them. In particular, a true tie does not invalidate the e-process framework or the FWER guarantee. Under a tie, the relevant pairwise hypothesis is simply harder to reject, so the corresponding e-process need not grow past the rejection threshold. More specifically, we may use $C_{j,k}=0$ to represent the case $p_{j,k}=1/2$, where $p_{j,k}$ denotes the pairwise winning probability defined on Line 155. The denominator of the mixture likelihood-ratio e-process in (1) is $\sup_{p\in[0,1/2]}L(p;S_t,t)$, which automatically includes the tie case $1/2$. Therefore, the proposed procedure effectively tests $H_0^{(j,k)}: C_{j,k}=\{0,1\}$ against $H_1^{(j,k)}: C_{j,k}=-1$.
>
> We agree that this point should be stated more explicitly in the paper. In the revision, we will clarify that ties are permitted under the sequential testing framework and that the mathematical formulation will explicitly include $C_{j,k}=0$ to denote the tie case.
>
> Key question 1(2): E-process runs forever if two models tie.
>
> Thank you for this important question. Our procedure addresses this issue automatically through the sequential testing mechanism itself, rather than a stopping rule. The logic behind the procedure is that when there are few pairwise comparison data, none of the hypotheses can be rejected because the available evidence is insufficient. The procedure treats all models as having comparable performance at that stage. If two models are truly tied, the corresponding e-process need not cross the threshold during the sampling. As a result, the pair simply remains unresolved, which is exactly the desired behavior: the procedure keeps treating the two models as indistinguishable.
>
> In the revision, we will clarify that when the available evidence is insufficient, the procedure treats the competing models as having the same performance.
>
> Key question 2(1): Transitivity and the pruning step.
>
> Response: Thank you for raising this important concern. We agree that strong non-transitivity may arise for some specific cases. We would like to clarify the following.
>
> First, transitivity is used only to support the pruning step and improve efficiency. If one does not wish to assume transitivity, the pruning step can simply be removed from the algorithm. SERPANT remains a valid multiple-testing procedure with rigorous FWER control. It evaluates each pair independently, outputting a directed graph of significant pairwise preferences. Any cycles (A$\to$B$\to$C$\to$A) are faithfully captured rather than forced into a hierarchy.
>
> Second, in many ranking settings, especially for tasks with scalar evaluation scores such as summarization, coding, or mathematics, transitivity is a natural and widely adopted structural assumption. More broadly, when users request a ranking from best to worst, this already assumes a transitive ordering.
>
> In the revision, we will make clear that transitivity is an optional structural assumption used only for pruning, not for testing directional hypotheses.
>
> Key question 2(2): Pruning step to multi-dimensional LLM skills?
>
> Thank you for this important question. Most real-world jobs are inherently multi-dimensional, so a single global ranking may be less informative than task-specific rankings. A natural approach is therefore to partition tasks into classes and report separate rankings. If a single overall ranking is desired, one can instead specify a target mixture over tasks, such as 50% math and 50% coding. Our procedure handles this directly by sampling from the chosen mixture and inferring from the resulting aggregate winning probabilities, thereby supporting both class-specific and weighted overall rankings.
>
> In addition, we use the Databricks setting to assess the benefit of pruning via transitivity and to examine whether the required condition holds under the same setup as in the main paper. As expected, disabling pruning results in some power loss, but we do not find any contradiction.
> | Dataset | Transitivity | t=1000 | t=2000 | t=3000 |
> |---|---|---:|---:|---:|
> | **Databricks** | Prune by transitivity | **28 (62%)** | **41 (91%)** | **43 (96%)** |
> |  | Without Prune | 24 (53%) | 35 (78%) | 38 (84%) |
>
> Key question 3: Threshold scale.
>
> Thank you for this important question. We agree that the threshold becomes more conservative as the number of models grows. The e-process is an exponential-evidence procedure. When two models have different behaviors, the corresponding e-value typically grows exponentially fast with the number of comparisons. As a result, the ordering can still be detected within a reasonable comparison budget. In the revision, we will make this clear.

---

> > ### Author Rebuttal · Reviewer_wgk6 · 2026-04-03
> >
> > I thank the authors for the detailed response and additional experiments. I will maintain my positive assessment of this paper. If accepted, I encourage the authors to incorporate the key clarifications and results presented in the rebuttal into the final version.

---

### Decision · Program_Chairs · 2026-04-30

**Decision:**

Accept (regular)

**Comment:**

This paper addresses the sequential ranking problem of LLMs that provide guarantees that are valid at any time. The authors proposed the SERPANT algorithm,  where the algorithm sequentially selects and samples a pair of the models and then update their e-values.  SERPANT also leverages the transitive structure for the pruning. The authors provide theoretical guarantees that familiy-wise error rate at any stoping time. The sampling rule is based on an adaptive sampling rule called Tournament Sampling, in which uncertainty and the signal strength are blended using the linear combinations.

The reviewers praised (i) the formulation with e-processes, as well as the theoretical guarantee (ii) the idea of tournament sampling rule, and (iii) the empirical strength of the algorithm. One of the primary concerns among the reviewers was whether the transitivity assumption is necessary for the algorithm and theory. The rebuttals clarified that transitivity is only used for pruning steps, and that, for domains where the transitivity assumption is unlikely to hold, these steps can simply be omitted. Following the discussion, we concluded that the authors could address this issue in the revised version.